# Comparative assessment of impact analysis methods applied to large commercial aircraft crash on reinforced concrete containment

**Muhammad Sadiq[1], Rao Arsalan Khushnood[1,2]\*, Muhammad Ilyas[3], Wasim Khaliq[1], Shaukat Ali Khan[4], Pan Rong[5]**

**1** National University of Sciences and Technology (NUST), Islamabad, Pakistan, **2** Politecnico Di Torino, Turin, Italy, **3** GIK Institute of Engineering Sciences and Technology, Topi, Pakistan, **4** Abasyn University, Peshawar, Pakistan, **5** Nuclear and Radiation Safety Center (NSC), Ministry of Environmental Protection (MEP), Beijing, China

\* rao.khushnood@polito.it, arsalan.khushnood@nice.nust.edu.pk

## Abstract

The precise evaluation of the potential damage caused by large commercial aircraft crash into civil structures, especially nuclear power plants (NPPs), has become essential design consideration. In this study, impact of Boeing 767 against rigid wall and outer containment building (reinforced concrete) of an NPP are simulated in ANSYS/LS-DYNA by using both force time history and missile target interaction methods with impact velocities ranging from 100 m/s to 150 m/s. The results show that impact loads, displacements, stresses for concrete and steel reinforcement, and damaged elements are higher in case of force time history method than missile target interaction method, making the former relatively conservative. It is observed that no perforation or scabbing takes place in case of 100 m/s impact speed, thus preventing any potential leakage. With full mass of Boeing 767 and impact velocity slightly above 100 m/s, the outer containment building can prevent local failure modes. At impact velocity higher than 120 m/s, scabbing and perforations are dominant. This concludes that in design and assessment of NPP structures against aircraft loadings, sufficient thickness or consideration of steel plates are essential to account for local failure modes and overall structural integrity. Furthermore, validation and application of detail 3D finite element and material models to full-scale impact analysis have been carried out to expand the existing database. In rigid wall impact analysis, the impact forces and impulses from FE analysis and Riera's method correspond well, which satisfies the recommendations of relevant standards and further ensure the accuracy of results in full-scale impact analysis. The methodology presented in this paper is extremely effective in simulating structural evaluation of full-scale aircraft impact on important facilities such as NPPs.

**Data Availability Statement:** All relevant data is included within the manuscript.

**Funding:** The research presented in this paper was supported by National University of Sciences and Technology (NUST), Islamabad, Pakistan and the Office of the Higher Education Commission grant no. 10232 to RAK. Any opinions, findings, and conclusions in this paper are those of authors and do not necessarily reflect the views of sponsors.

**Competing interests:** The authors have declared that no competing interests exist.

# Introduction

Containment structures for nuclear power plants (NPPs) are designed and constructed to prevent unacceptable releases of radioactivity during an accident. In case of leakage after failure of containment, consequences are serious. Therefore, safety assessment against large commercial aircraft impact is required to be performed during the design of NPP structures [1]. The concept of redundancy is included in design of NPP to ensure fundamental nuclear safety functions, like cooling of core and confinement of containment against design basis accident conditions. These functions are required to be maintained especially in case of large commercial/military aircraft impact. For calculation of aircraft impact load, Riera loading function [2] was used in numerous studies and validated by experimental results reported by Sugano, Tsubota [3]. Depending on data availability and the expected analysis results, two methods for aircraft impact can be used as per Nuclear Energy Institute's (NEI) 07–13 [4] (i) the force time-history method by Riera [2] is applied to an analytical model of the structure (outer containment building in this case) in a similar manner like time history analysis, and (ii) missile-target interaction analysis method, in which analysis models of aircraft and target structure are built together, and response against the aircraft impact is calculated as an initial velocity problem. The developed nonlinear models have more number of elements and complicated as compared to force time-history analysis method. Consequently, for this method, more detailed data regarding mass and stiffness of aircraft is required than the time history analysis method but can possibly give precise results [4]. On the other hand, force time-history method gives conservative, but reasonably accurate results as concluded in previous studies [5, 6]. Also, this method has numerical limitations as it does not consider the interaction between aircraft & target structure, leading to unavailability of results in terms of residual velocities and fracture process. It can only be applied normal at the target and is difficult to determine loading area. However, missile-target interaction analysis method can overcome these limitations.

Many analytical studies have been performed regarding impact of large commercial aircrafts by using both force time history and missile-target interaction methods [5, 7–15]. Abbas, Paul [7], evaluated the impact of aircraft crash against an outer containment of an NPP by using the Riera loading function at different locations. They pointed out that horizontal impact of an aircraft adjacent to junction of hemispherical dome and cylindrical wall is the most critical location for aircraft impact. Wierzbicki and Teng [8], simulated the collapse of aircraft subjected to World Trade Center in the "September 11" event by using crash site data. Also, Karim and Fatt [9] simulated the impact of Boeing 767 against the outer columns of the World Trade Center by using LS-DYNA and determined that minimum impact velocity would be 130 m/s to just penetrate the outer columns. Arros and Doumbalski [5] performed the impact analysis of an aircraft similar to Boeing 747 to nuclear building by using both methods of impact analysis. They concluded that Riera time history method provides conservative results which are also sensitive to the associated assumptions of loading area and time of impact load application. Kostov, Henkel [12], performed detailed impact analyses of large aircraft against a reactor building by changing both the locations of impact and loading intensity. In their study, equivalent impact loadings are characterized by load time functions determined by three different methods. These include the loading calculated by Riera's method, load time function calculated by finite element (FE) analysis and coupled dynamic analysis due to contact between target and projectile (also called missile target interaction method). More recently, Lu, Lin [13] evaluated the impact of three Boeing 767 finite element models with dissimilar fidelities against the rigid target and containment building by using LS-DYNA. The simulation results indicated that impact force time histories and impulses vary significantly among the three considered FE models for the low velocity impact case. These authors also

recommended to consider the internal structures of an aircraft in impact analysis for more accurate results. By review of said studies, it is noted that mostly, simplified finite element (FE) aircraft models based on available data were created, which may not simulate the actual impact scenario of aircraft against its target structure. The lack of validation of input parameters like material models, contact algorithms etc. in some published studies raises concerns on the accuracy of results. The recommendation of NEI 07–13 [4] regarding demonstration of defined aircraft model against the impact of rigid wall requires implementation at larger scale. Moreover, few studies were performed at different impact velocities to compare the results of both impact analysis methods. In current study, material models and contact algorithms validated by author in previously study [16] are adopted for accurate results. The defined FE model of aircraft is verified for its accuracy by rigid wall impact analysis as per NEI 07–13 [4].

The current impact analysis focuses on impact responses of the outer containment building against the large commercial aircraft similar to Boeing 767. In order to verify the accurate mass and stiffness distribution of defined FE model of aircraft, rigid wall impact analysis is done for two different velocities of aircraft i.e. 100 m/s and 150 m/s as per recommendation of NEI 07–13 [4] and the results of impact forces are compared with those of Riera's curve. Then full-scale impact analysis of aircraft is performed against the outer containment building (reinforced concrete) in LS-DYNA with both force time history and missile target interaction methods. Also, the applicability range of material models especially the Winfrith concrete model (*MAT_084) from simulation of scale model tests to full-scale impact analysis is investigated. The study highlights the comparison of two impact analysis methods, the behavior of aircraft and outer reinforced concrete containment building and availability of important aircraft data such as initial impact velocity and mass distribution in determining local failure modes and overall structural response. Such failure modes are outlined in NEI 07–13 [4].

## Numerical modeling of aircraft and outer containment building

**Aircraft modeling.** A typical large commercial aircraft, Boeing 767 is simulated in ANSYS based on open source data and published work [17]. This aircraft was also impacted the World Trade Center in September 11, 2001 event. Various relevant research work have been performed using this aircraft e.g. [8, 13], and considerable data are available in the published studies. Geometric parameters of this aircraft are shown in Fig 1(A). Impact analysis requires structural details of Boeing 767 and its mass distribution. From reference [17], the total mass of the aircraft is taken as 179,330 kg (179 tons). The aircraft is assumed to be composed of three types of structures namely fuselage, rigid engines, and strong but crushable wings. The inner details of engines are not available, which may affect the crushing of engine during the impact. The model of aircraft could be further refined on the availability of detailed information of engine core. The thicknesses of fuselage, wings and engine are 0.0184 m, 0.0345 m and 0.0010 m, respectively. These thicknesses are calculated as per equations given in reference [17]. The fuel tank is located in the wings based on published studies. The fuselage of aircraft usually comprised of rings and stringers attached to sheet metal. These inner parts are converted into an equivalent thickness which preserves the same mass as that of actual fuselage.

In the finite element model, the equipment loads of major Boeing 767 parts are converted to masses along the position of aircraft as per their mass distribution in actual aircraft [13] as shown in Fig 1(B). Furthermore, geometry of aircraft created in ANSYS APDL and FE mesh are shown in Fig 2. All three types of structures (fuselage, wings, and engine) are modeled by using the shell element (SHELL163) with 2024-T351 aluminum. The mesh size in current full-scale impact analysis is larger than adopted in simulation analysis of 1/7.5 scale model tests

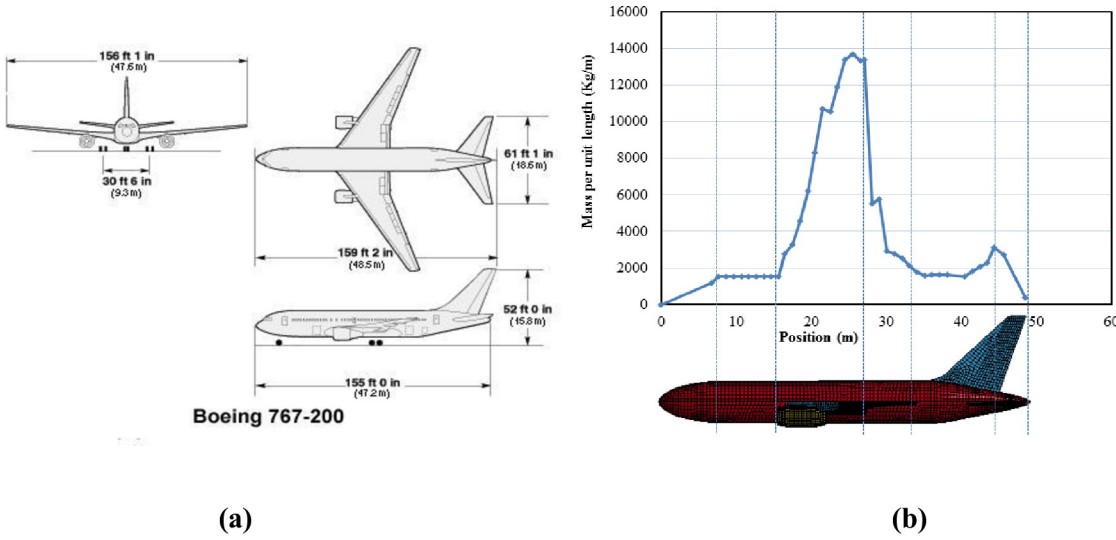

**(a)**　　　　　　　　　　　　　　　　　　　　**(b)**

**Fig 1. Dimensions and mass distribution of Boeing 767.** (a) Dimensional parameters. (b) Longitudinal mass distribution.

[16] and its effects are checked by performing various analysis runs of defined aircraft model on rigid wall. The time histories of the impact forces were compared with Riera curves. Most comparable force and impulse time histories were obtained once the mesh size of 400 mm for fuselage & wings and 200 mm for engines were selected in aircraft modelling. Further refinement of mesh sizes did not lead to significant improvement to the solution. Therefore, mesh sizes of 400 mm for fuselage & wings and 200 mm for engines are chosen in the current analysis as shown in Fig 2(B).

Material model, *MAT_PLASTIC_KINEMATIC (*MAT_003) available in LS-DYNA [18] is used, after validation of constitutive models with test results by author [16] and the input parameters are mentioned in Table 1. Fuel is modeled by solid elements with material model (*MAT_NULL) available in LS-DYNA. This material model also requires an equation of state. Therefore, *EOS_GRUNEISEN equation of state [18] is employed in this study. In LS-DYNA,

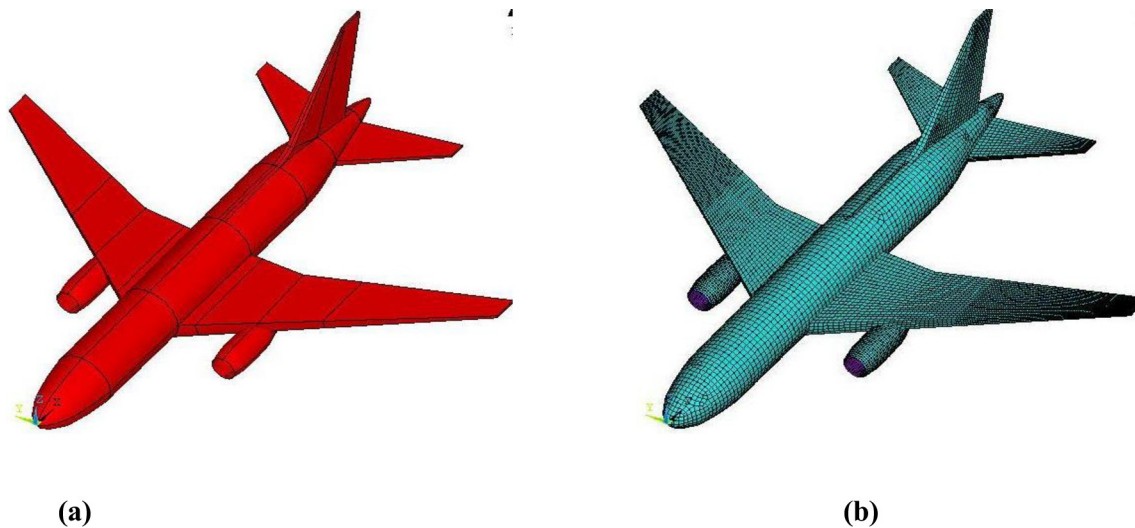

**(a)**　　　　　　　　　　　　　　　　　　　　**(b)**

**Fig 2. Finite element model of Boeing 767.** (a) Unmeshed geometry. (b) Meshed model.

Table 1. Input material properties of aircraft parts for LS DYNA material model.

| MAT_003 | Fuselage | Wings | Engine |
|---|---|---|---|
| Mass density, RO (kg/m$^3$) | 4810 | 1030 | 8.16 x 10$^4$ |
| Young's modulus, E (N/m$^2$) | 7.31 x 10$^{10}$ | 7.31 x 10$^{10}$ | 7.31 x 10$^{10}$ |
| Poisson's ratio, PR | 0.33 | 0.33 | 0.33 |
| Yield stress, SIGY (N/m$^2$) | 3.24 x 10$^8$ | 3.24 x 10$^8$ | 3.24 x 10$^8$ |
| Failure strain, FS (m/m) | 0.1 | 0.1 | 0.1 |
| MAT_NULL (Fuel) | | | |
| Mass density, RO (kg/m$^3$) | 512 | | |
| Pressure cutoff, PC (Pa) | -1 | | |
| Relative Volume, TEROD | 2 | | |
| Relative Volume, CEROD | 0.5 | | |
| EOS_GRUNEISEN | | | |
| Gruneisen Constant, C (m/s) | 1560 | | |
| Coefficient of slope, S$_1$ | 1.34 | | |
| Gruneisen gamma, γ | 2.0 | | |

the input parameters shown in Table 1 for *MAT_NULL and *EOS_GRUNEISEN are set by using the recommendations given in LS-DYNA keyword manual [18] and review of similar published studies, e.g. [6, 17]. Hence, this material model uses a material having no yield strength and possess fluid like characteristics. It is mentioned that dynamic behavior of the target structure against the post thermal effect are not evaluated in current study. This could be done by considering the appropriate material model available in LS-DYNA.

**Modeling of outer containment building.** In design of NPPs, the main function of outer containment is to protect the inner containment against external events. The outer containment building considered in this study is a typical reinforced concrete structure adopted against the impact of large commercial aircrafts in newly designed reactors. FE model of outer containment building is built with concrete and reinforcement bars (horizontal and vertical bars) as shown in Fig 3. Concrete is modeled by using element type SOLID164 with mesh size of 0.6 m, while steel bars are embedded as discrete bars by using the BEAM161 element available in ANSYS/LS-DYNA [19]. The equivalent diameters for horizontal and vertical steel bars are 63.2 mm and 58.3 mm, respectively. The total reinforcement ratio is 2.4%. The coupling between these reinforcing beams and solid concrete elements is ensured by using *CONTRAINED_LAGRANGE_IN_SOLID feature available in LS-DYNA [18]. While modelling, the bottom nodes of all elements (concrete and rebars) are constrained for both translations and rotations degree of freedom to achieve fixed boundary conditions. The main objective of simulation analysis performed by author in 2014 [16] was the validation of constitutive models with test results for full scale impact analysis against large commercial aircraft. Based on that study, two material models, Winfrith concrete model (*MAT_084) and Continuous Surface Cap Model (CSCM) model (*MAT_159) were used. Since the results of Winfrith concrete model (*MAT_084) corresponded well with the experimental results [16], hence the same model is selected for concrete of outer containment building in this study with the input parameters shown in Table 2. Similarly, for rebars, material model *MAT_PLASTIC_KINE MATIC (*MAT_003) is used as shown in Table 2. A brief description of both material models of concrete and rebars is given in references [18, 20]. While, the details of failure criteria associated with both material models is mentioned in author's prior study [16]. Moreover, in aircraft crash simulation, effect of damping is neglected because it does not affect the response against impulse loadings [21].

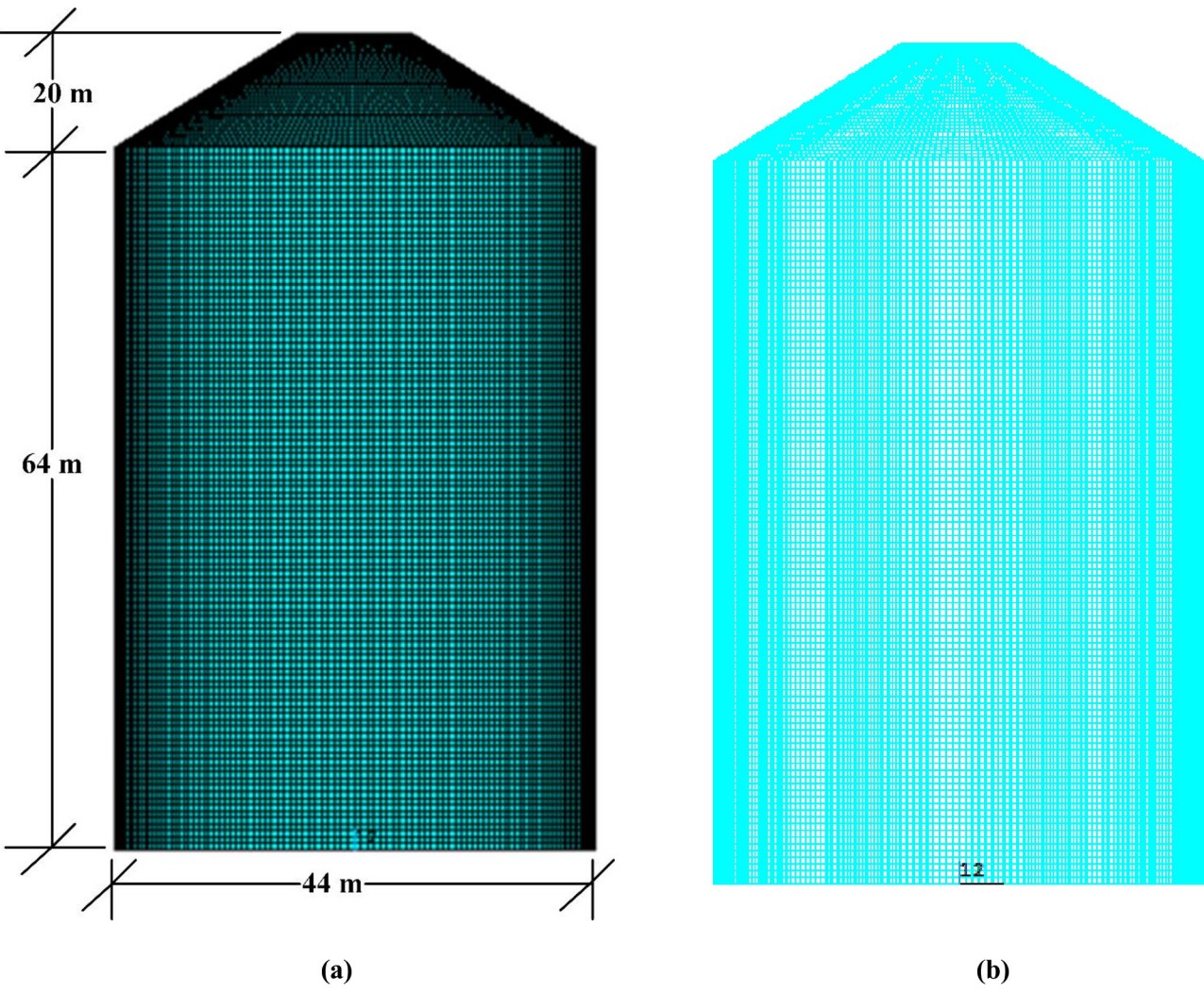

**Fig 3. FE model of outer reinforced concrete containment building.** (a) Concrete modeled with SOLID164. (b) Reinforcement bars in vertical and hoop directions.

## Methods of impact analysis

According to NEI 07–13 [4], the following two methods may be adopted for impact evaluation based on data availability and the expected analysis results. In the current study, both methods are used for full scale impact analysis.

### Force time-history analysis method

The force time history analysis is also called as the Riera methodology and its loading function is approved, as mentioned in US Department of Energy (DOE) STD-3014 [22]. Fig 4 shows the Riera model assuming a very thin "deformation" zone adjacent to the rigid target and a rigid zone within a control volume. The basic assumptions of the Riera method are [4] as follows:

1. It is assumed that missile/aircraft is impacted on rigid target

2. The longitudinal axis of the aircraft or missile is oriented normal to the target

**Table 2. Input material properties for LS DYNA material models.**

| MAT_084 (Concrete) | |
| --- | --- |
| Mass density, RO (kg/m$^3$) | 2500 |
| Initial tangent modulus, TM (N/m$^2$) | 3.45 x 10$^{10}$ |
| Poisson's ratio, PR | 0.2 |
| Uniaxial compressive strength, UCS (MPa) | 83.5 |
| Uniaxial tensile strength, UTS (MPa) | 6.18 |
| Fracture energy, FE (N.m/m$^2$) | 100 |
| Aggregate radius, ASIZE (m) | 0.01 |
| Rate effects (0 = ON, 1 = OFF) | 0 |
| **Input for MAT_ADD_EROSION (concrete)** | |
| Max principal strain at failure, MXEPS | 0.05 |
| **MAT_003 (Rebars)** | |
| Mass density, RO (kg/m$^3$) | 7800 |
| Young's modulus, E (N/m$^2$) | 2.0 x 10$^{11}$ |
| Poisson's ratio, PR | 0.3 |
| Yield stress, SIGY (N/m$^2$) | 5.74 x 10$^8$ |
| Failure strain, FS (m/m) | 0.1 |

3. The aircraft is divided into two portions: one is rigid portion moving with velocity (V) and the second portion is crushed and resulted into zero velocity

4. All crushing of aircraft/missile takes place within a local portion located near to rigid target

5. The material behavior of the airframe is observed as ideal plastic

For aircraft hitting a target perpendicular to the impact surface at time $t$, the impact force is given by following equation, also called Riera function [2].

$$F(t) = Pc[x(t)] + \mu[x(t)]v^2(t) \tag{1}$$

Where $x(t)$ is the distance taken from the nose of the aircraft to present location, $Pc[x(t)]$ is crushing strength of the aircraft, $\mu[x(t)]$ is the mass per unit length of the aircraft as shown in Fig 1(B), and $v(t)$ is the velocity of the aircraft which has not been crushed. Both $Pc[x(t)]$ and $\mu[x(t)]$ are measured along position of aircraft, which is generally taken from the nose. Sugano, Tsubota [3] validated the Riera methodology against full-scale test data obtained from impact of F-4 Phantom military aircraft on rigid target, and determined the coefficient α as 0.9. Resultantly, the Eq (1) is modified with α as given below, which can be used to determine the force time history.

$$F(t) = Pc[x(t)] + \alpha.\mu[x(t)]v^2(t) \tag{2}$$

The coefficient α is highly dependent upon the characteristics of mass distribution of each aircraft and difficult to determine experimentally [23]. In current study, the influence of the crushing strength on the impact forces and impulses are evaluated by using the Riera equation [2]. The actual crushing strength of Boeing 707–320, as adopted in reference [24] with four different percentages, is considered in calculation of impact forces and impulses. The calculations are performed for two impact velocities of 100 m/s and 150 m/s as show in Figs 5 and 6. For both cases, the impact force and impulse time history curves are very similar. At impact velocity of 100 m/s shown in Fig 5, maximum variation of 1.16% in impact forces and 5.50% in

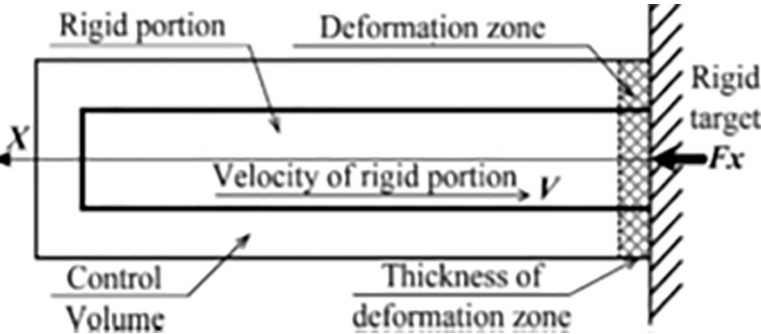

**Fig 4. Aircraft striking a target as per Riera model [14].**

impulses are observed due to four varying crushing strengths. As shown in Fig 6, the variation in impact forces and impulses are further reduced to 0.40% and 2% at higher impact velocity of 150 m/s. It is observed that these small variations in impact forces and impulses due to varying crushing strengths would have insignificant influence. Mainly, mass and velocity of the aircraft dominate/control the force-time history and are referred to as controlling cases. Since the crushing strength of the Boeing 767 aircraft model is not available in open literature, therefore, it is acceptable approach to adopt the crushing strength of available Boeing 707–320 from reference [24]. T. Zhang et al [14] also utilized the crushing strength of Boeing 707–320 in impact simulation of A320 aircraft after determining its influence on impact forces and impulses by similar approach.

$\mu[x(t)]$ is calculated by dividing the aircraft into small portions (approximately 1 m long) along the length. By knowing the $Pc[x(t)]$, $\mu[x(t)]$ and the initial impact velocity $\upsilon$, a spreadsheet program is developed to calculate the impact force as a function of time with two velocities. The Riera's force time history curves, corresponding to two velocities of 100 m/s and 150 m/s, are applied to a zone of outer containment building as highlighted in Fig 7.

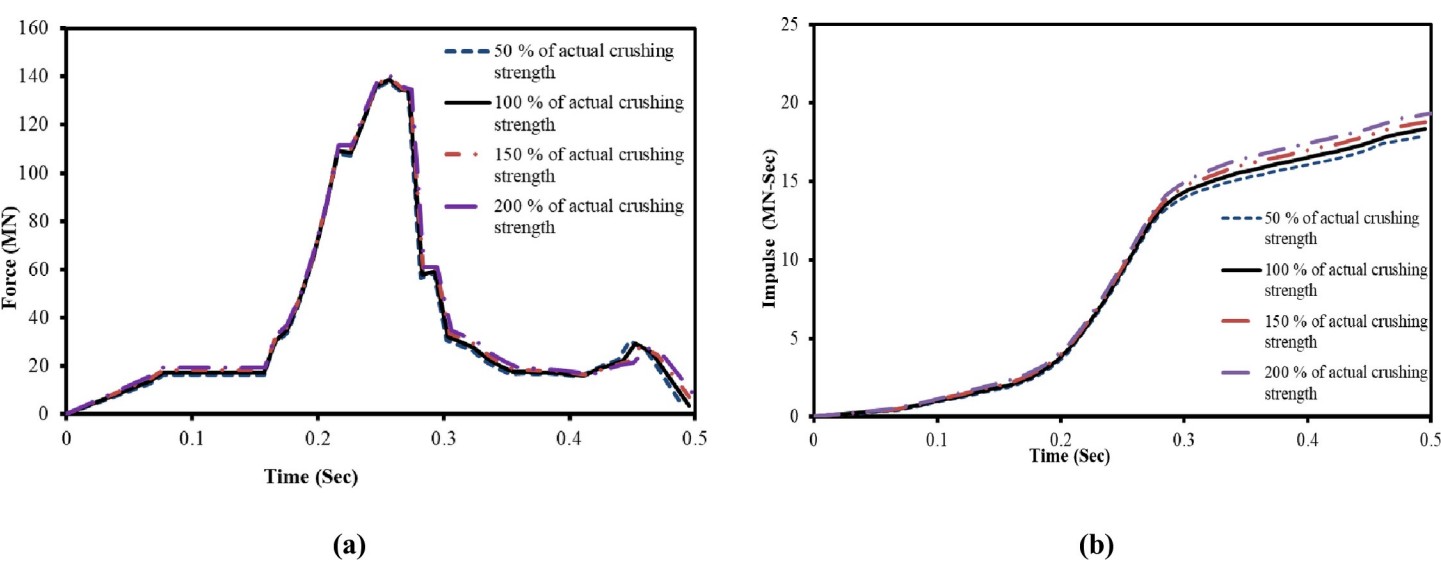

**(a)**                                                                                          **(b)**

**Fig 5. Influences of aircraft crushing strength on impact forces and impulses at 100 m/s impact velocity.** (a) Impact force time histories. (b) Impulse time histories.

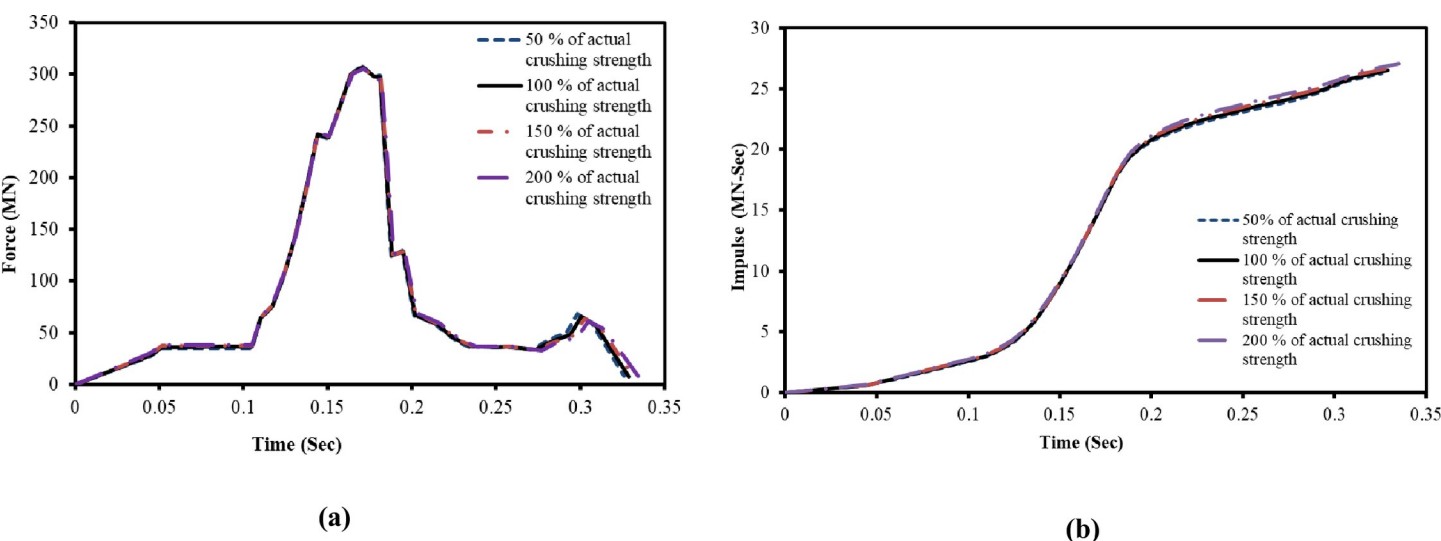

**Fig 6. Influences of aircraft crushing strength on impact forces and impulses at 150 m/s impact velocity.** (a) Impact force time histories. (b) Impulse time histories.

## Missile-target interaction analysis method

In this method, analysis models of aircraft and target structure are built together, and response against the aircraft impact is calculated as initial velocity problem [4]. Further, in this method, aircraft can be oriented at an arbitrary/desired angle. However, in the present paper, a normal impact with aircraft crashing from the front, as shown in Fig 8, is selected as it corresponds to the most unfavorable condition giving the maximum structural response in this condition [25].

## Rigid wall impact analysis

According to NEI 07–13 [4], a demonstration that defined aircraft analysis models (Boeing 767 FE model in current study) properly represents the Riera force-time history is needed. This is done by applying the aircraft model at a defined initial velocity, to a rigid target or unyielding surface. In this way, all of the impact energy is absorbed in aircraft deformation. In previous studies by Arros and Doumbalski [5], Wilt, Chowdhury [6], Siefert and Henkel [26] and Rashid, James [27], impact analysis on rigid wall was performed and the results were compared with calculated Riera's force time history. Accordingly, in current analysis, impact forces are calculated by two methods i.e., (i) the Riera method in a spreadsheet program with both $\alpha$ = 1 and $\alpha$ = 0.9, and (ii) impact of FE model of aircraft on rigid wall in LS-DYNA as illustrated in Fig 9. The impact forces are calculated for two impact velocities i.e., 100 m/s and 150 m/s. Lu, Lin [13] also adopted 100 m/s and 150 m/s impact velocities for Boeing 767 in their study.

The fracture process in Fig 10 shows that aircraft is completely destroyed after impacting the rigid wall at 100 m/s impact velocity. The comparison of calculated forces in LS-DYNA is made with forces calculated by Riera's force time history with both $\alpha$ = 1 and $\alpha$ = 0.9, as shown in Fig 11, for two impact velocities, respectively. The total impulse carried to the rigid target is calculated by integrating the force vs time curves as shown in Fig 12. The impact forces and impulses for both impact velocities are summarized in Table 3. At 100 m/s, maximum force of 140.7 MN is observed at 0.23 sec for impact analysis of FE model of aircraft against rigid wall in LS DYNA run. The corresponding maximum forces of 138.58 MN and 125.41 MN are found for $\alpha$ = 1 and $\alpha$ = 0.9 respectively in case of Riera method. Impact duration of 150 m/s

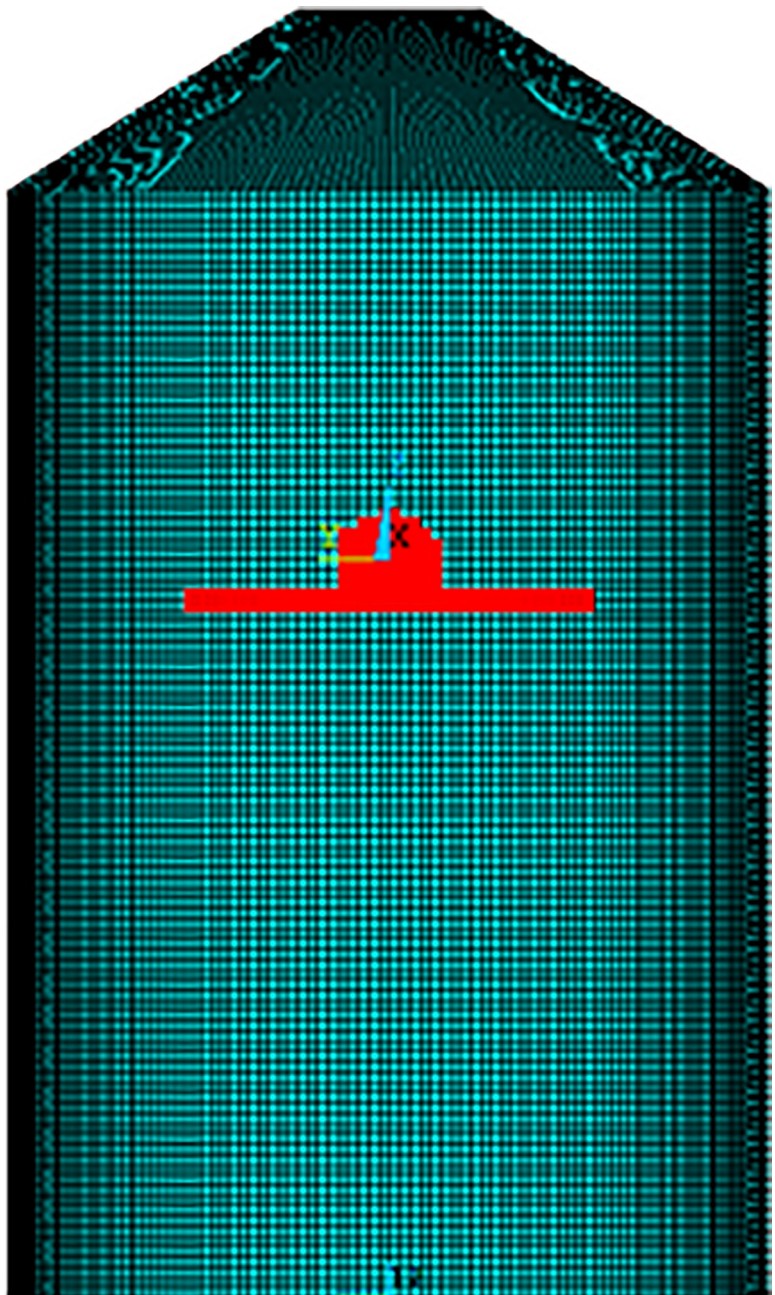

**Fig 7. FE model of outer containment building showing loading areas for Riera force history.**

impact velocity case is shorter, but impact forces and impulses are more due to higher velocity of aircraft, i.e. $v^2(t)$ in Eqs (1) and (2). Similarly, Liu et al [15] found that with increasing impact velocity, the duration of the total impact force reduces but the peak value of the total impact force increases significantly. This indicates that velocity is the main factor determining the impact duration as the length of the aircraft is same during both impact velocities i.e. 100 m/s and 150 m/s.

As shown in Table 3 and Fig 11(B), at 150 m/s, maximum force of 308.83 MN at 0.15 sec is calculated in LS DYNA run. However, maximum forces of 307 MN and 277 MN at 0.17 sec for

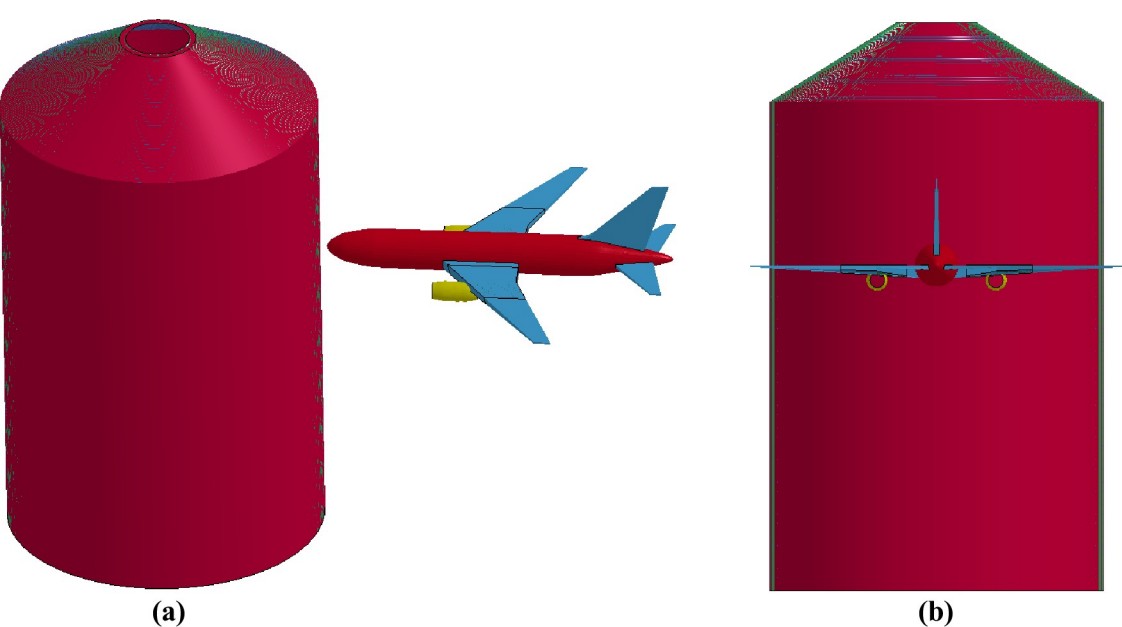

**Fig 8. FE model of aircraft and outer containment building for missile-target interaction method.** (a) Projection view. (b) Front view.

$\alpha = 1$ and $\alpha = 0.9$, respectively in case of Riera method. It is found that total impulses i.e. area under the curve in Riera method is overestimated by 14% for impact velocity of 100 m/s when $\alpha = 1$ and 6% with $\alpha = 0.9$. Similarly, the impulses for $\alpha = 1$ and $\alpha = 0.9$ are 15% and 6.4% higher when the impact velocity was 150 m/s. Zhang, Wu [14] also determined the impact

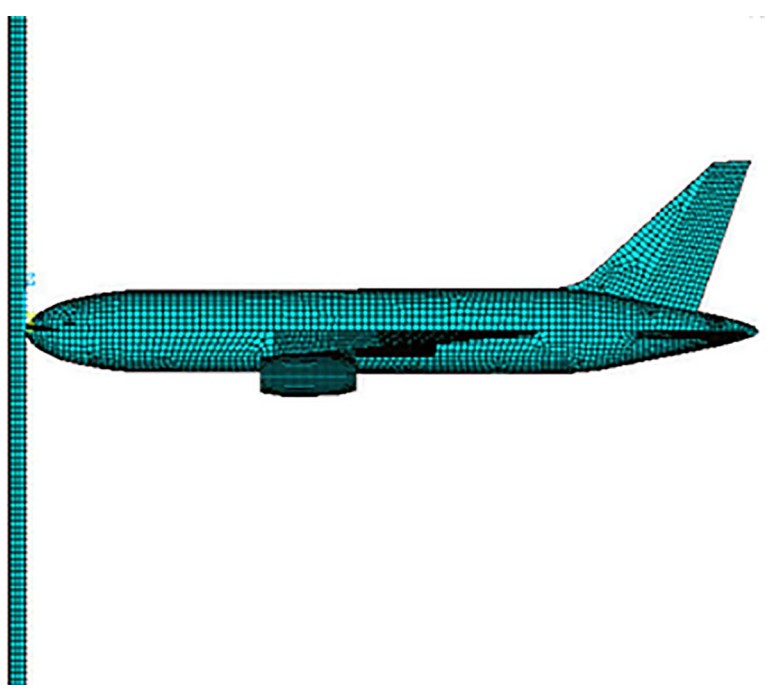

**Fig 9. Aircraft impact at rigid wall.**

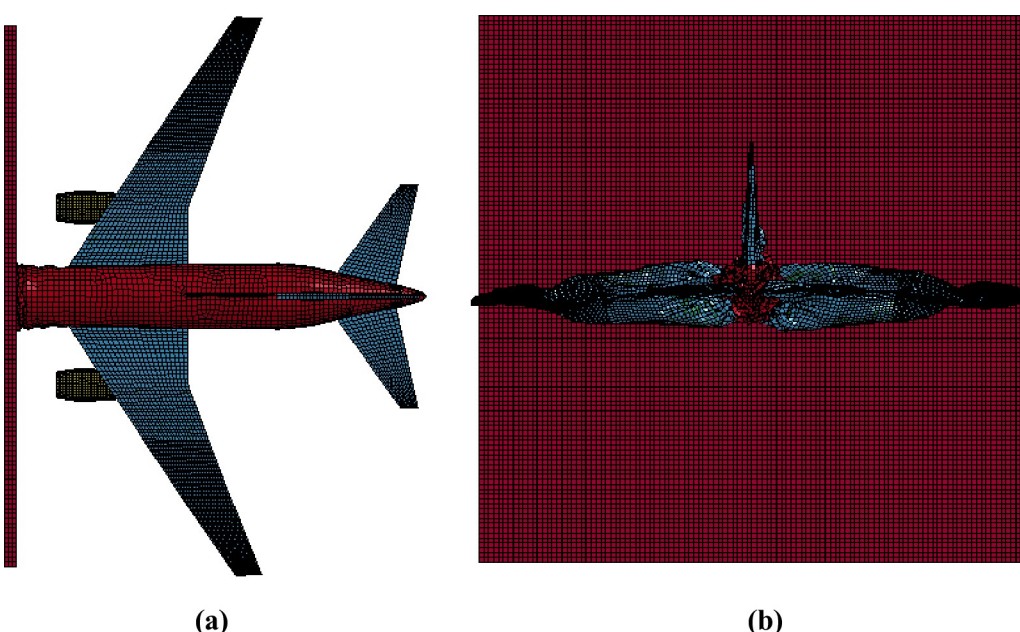

**Fig 10. Fracture process of Boeing 767 at rigid wall.** (a) Fracture process at 0.12 sec. (b) Destroyed aircraft at end of analysis time, i.e. 0.45 sec.

forces for A320 aircraft with α = 1 and α = 0.9 and concluded that impact force and impulse corresponded well with the results based on Eq (2) when α = 0.9. The difference observed in current rigid wall impact analysis is quite reasonable and could further be reduced by modeling the internal structures of the aircraft as pointed out by Lu, Lin [13]. These results show that FE model of Boeing 767 has almost similar characteristic to actual aircraft and meets the recommendation set forth in NEI 07–13 [4]. The results also illustrate that material models used for aircraft parts are reasonable. This provides necessary confidence for conducting full scale aircraft impact analysis.

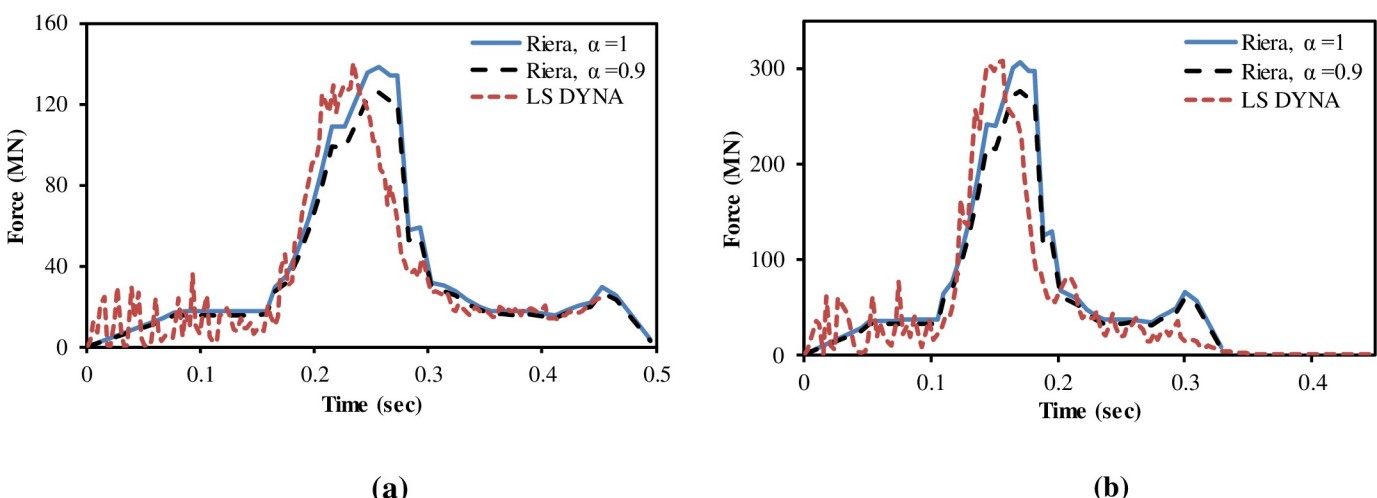

**Fig 11. Comparison of impact forces between Riera method and LS-DYNA run.** (a) 100 m/s impact velocity. (b) 150 m/s impact velocity.

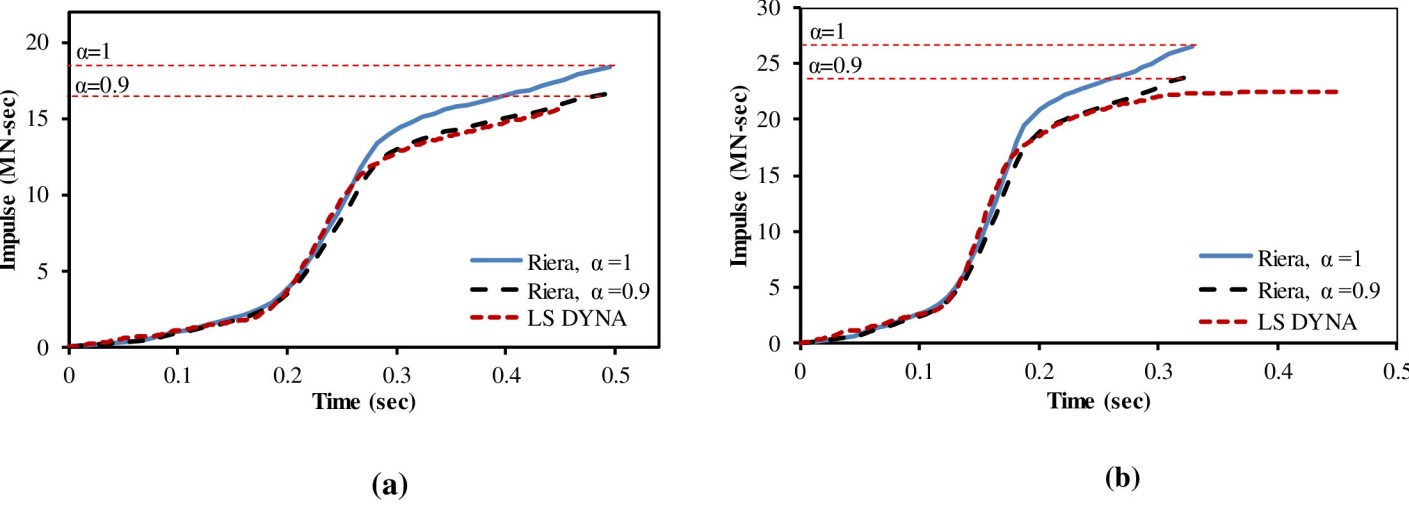

**Fig 12. Comparison of impulses between Riera method and LS-DYNA run.** (a) 100 m/s impact velocity. (b) 150 m/s impact velocity.

## Results and discussions of full-scale impact analysis

### Missile target interaction analysis method

The results of full-scale impact analysis of Boeing 767 aircraft model impacting the outer reinforced concrete containment structure are presented. For impact velocity of 100 m/s, fracture process of aircraft on outer containment building is shown in Fig 13, in which aircraft is almost completely destroyed and no perforation or scabbing occurred. According to NEI 07–13 [4], scabbing is defined as removal of material from the back face of the target structure (i.e., opposite direction to aircraft impact), while perforation is termed as the full penetration and passage of the aircraft through the target. These two types of local structural failure modes are required to be assessed for containment structures. From the residual velocities of uncrushed aircraft parts in Fig 14, it is shown that velocities of aircraft uncrushed parts decrease gradually and at 0.33 sec, becomes almost zero. The residual velocities of uncrushed aircraft parts are obtained by using *DATABASE_BINARY_D3THDT option available in LS-DYNA [18]. At defined output time intervals (e.g., DT = 0.03 sec in current analysis), velocity time history file is created for all defined parts of aircraft. The residual velocity time histories extracted from this file are plotted for individual parts (such as fuselage, wings and engine) as shown in Figs 14 and 17. According to NEI 07–13 [4], the term "residual velocity" is the exit velocity of missile/aircraft that has an initial velocity greater than the perforation velocity. In experimental studies such as performed by Jun Mizuno et al [28], the residual velocity times histories of uncrushed aircraft parts are obtained by means of on-board accelerometers and

**Table 3. Summary comparison of LS-DYNA and Riera impact loads.**

| Impact velocity | Methods | Max value of force (MN) | Impulse (MN-sec) |
|---|---|---|---|
| 100 m/s | LS DYNA | 140.7 | 15.62 |
| | Riera for α = 1 | 138.58 | 18.35 |
| | Riera for α = 0.9 | 125.41 | 16.66 |
| 150 m/s | LS DYNA | 308.83 | 22.45 |
| | Riera for α = 1 | 307 | 26.51 |
| | Riera for α = 0.9 | 277 | 23.98 |

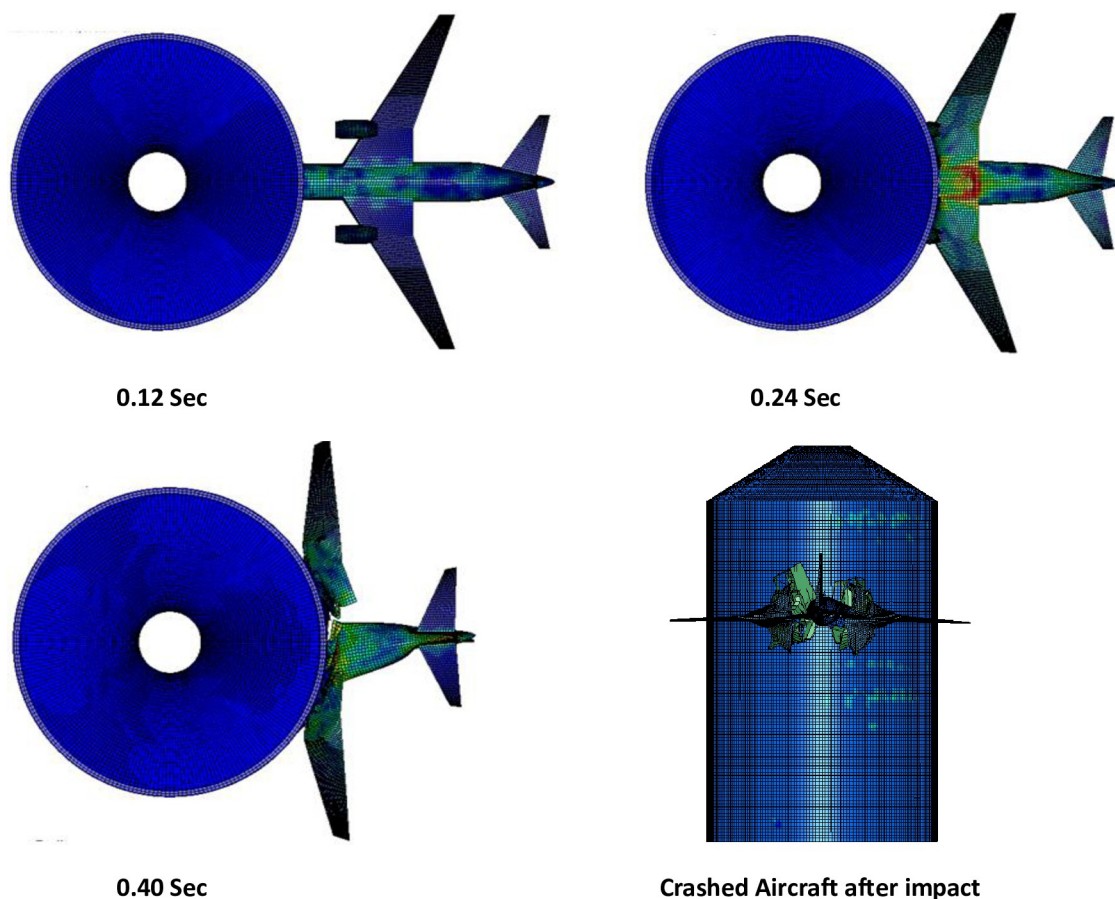

**0.12 Sec**

**0.24 Sec**

**0.40 Sec**

**Crashed Aircraft after impact**

**Fig 13. Fracture process at 100 m/s impact speed.**

high-speed video cameras. Displacement of a node (node # 24084) belonging to impact location, as shown in Fig 7, was extracted and is shown in Fig 15. It can be observed that displacement of this node increases when the nose of the plane encounters the reinforced concrete structure, i.e. time 0–0.1 sec. This is followed by a slight decrease when the cylindrical portion of fuselage is being crushed, i.e. time 0.1–0.2 sec. The displacement once again starts increasing when the wings and fuselage are in contact with the impact location. The displacement stays in a stable zone, maximum value of 0.082 m, during the crushing of wings, i.e. 0.2–0.36 sec. After the complete destruction of the wings, the node starts to come towards its original position.

In full-scale aircraft impact analysis at 150 m/s case, fracture process of aircraft and outer containment building is shown in Fig 16. There is no significant damage observed to containment building till first 0.12 sec. After this, relatively stiffer parts of the aircraft i.e. wings and engine come into contact and cause perforation. Front portions of fuselage and wings are destroyed while rear debris perforate the outer containment building. The steel bars and concrete fail when reach their predefined failure strains. The uncrushed parts still have significant kinetic energy and attain the residual velocities highlighted in Fig 17 at the end of analysis time i.e. 0.45 sec. The residual velocity of engine is more than wings and fuselage due to higher strength and stiffness. These residual velocities can further damage the components/systems. Meanwhile, outer and inner containments are designed with a reasonable gap (annular space) and important safety-related components/systems are located in inner containment. The

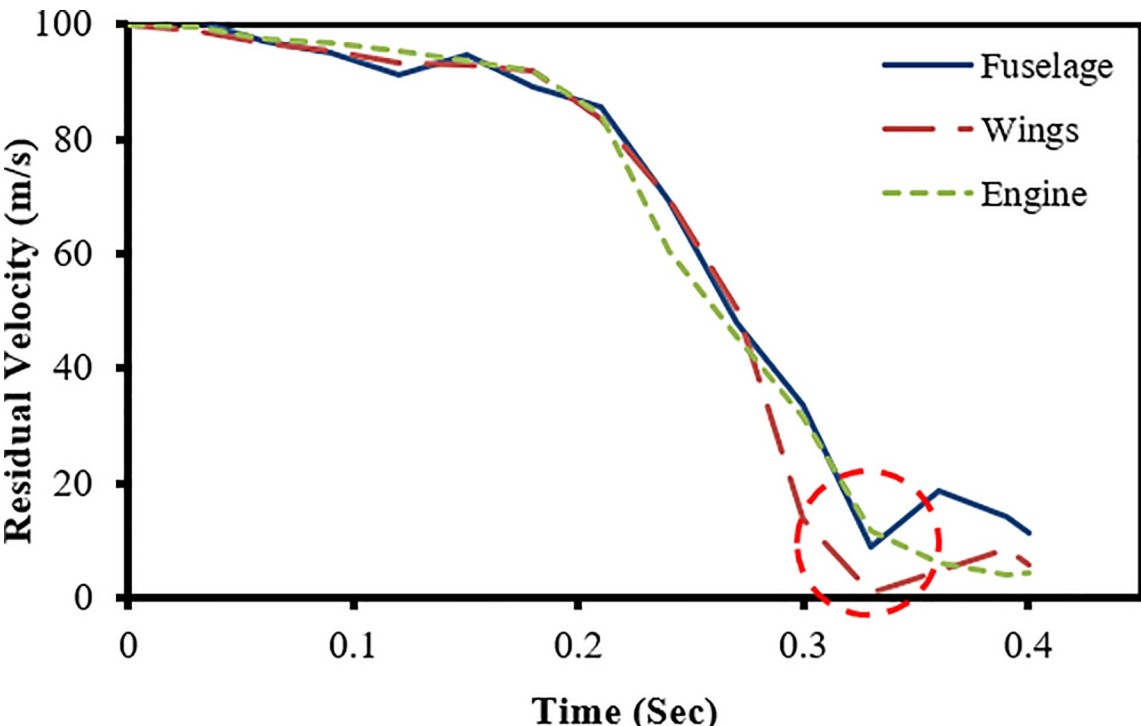

**Fig 14. Residual velocity of uncrushed aircraft parts against 100 m/s impact speed of aircraft.**

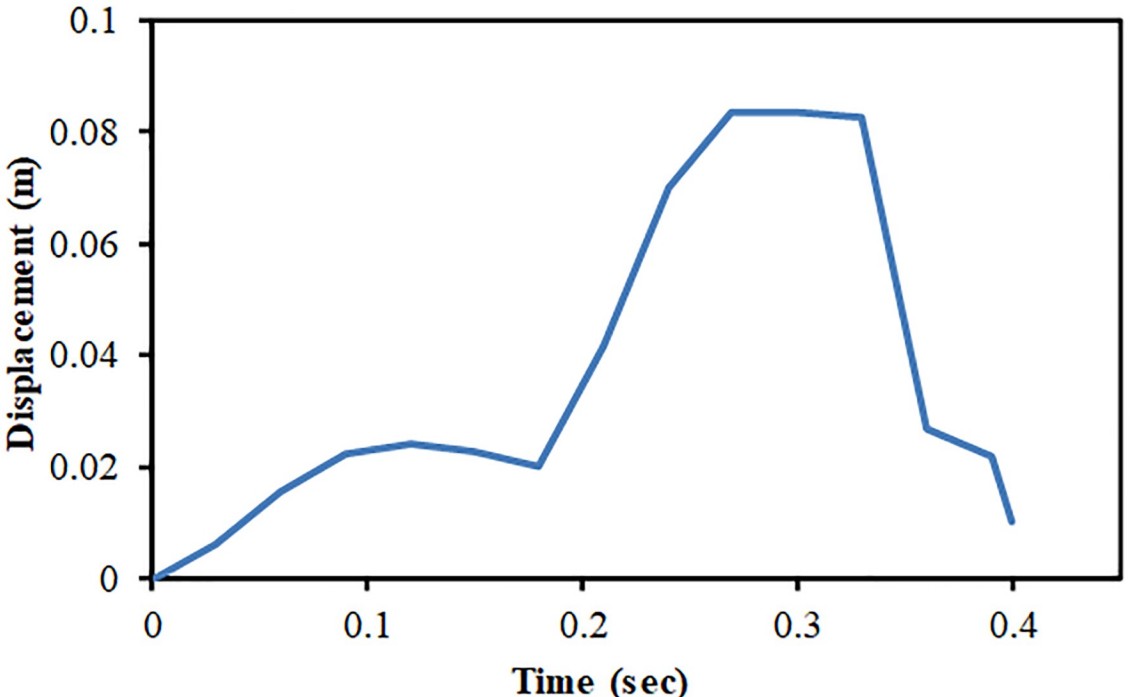

**Fig 15. Nodal displacement time history for impact speed at 100 m/s.**

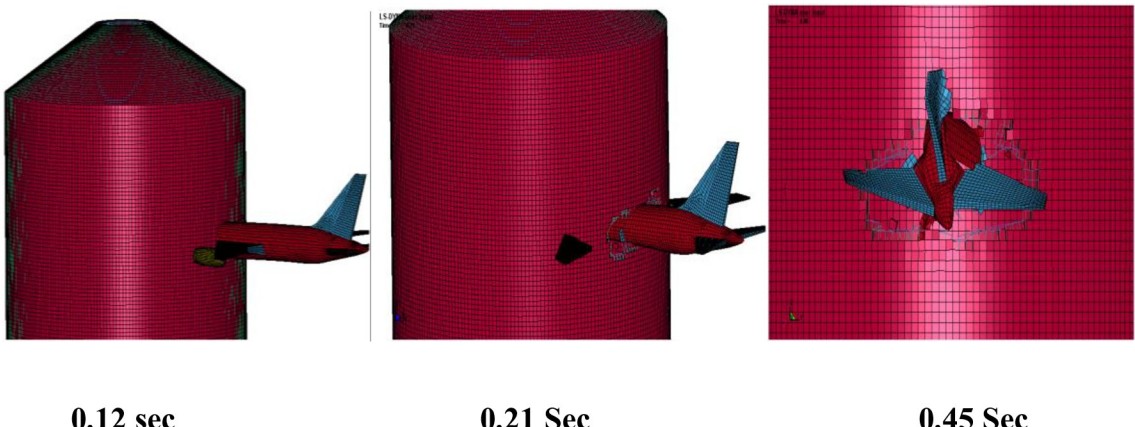

| 0.12 sec | 0.21 Sec | 0.45 Sec |

**Fig 16. Fracture process of aircraft and outer containment building at 150 m/s impact velocity.**

penetrations passing through annular space are protected with special guard pipes. These design measures are intended to prevent potential failures caused by perforation of aircraft. As a result of perforation at impact area, the maximum displacement of a node 49625 at impact area is more than 4 m as shown in Fig 18. As part of sensitivity study, the impact analysis is also performed at impact velocity of 120 m/s and it is observed that perforation of aircraft parts could have been prevented if the thickness of outer containment building is increased. Another possibility is the provision of steel plate which is very useful in resisting the perforation and scabbing of concrete as reported in actual experiments performed by Mizuno,

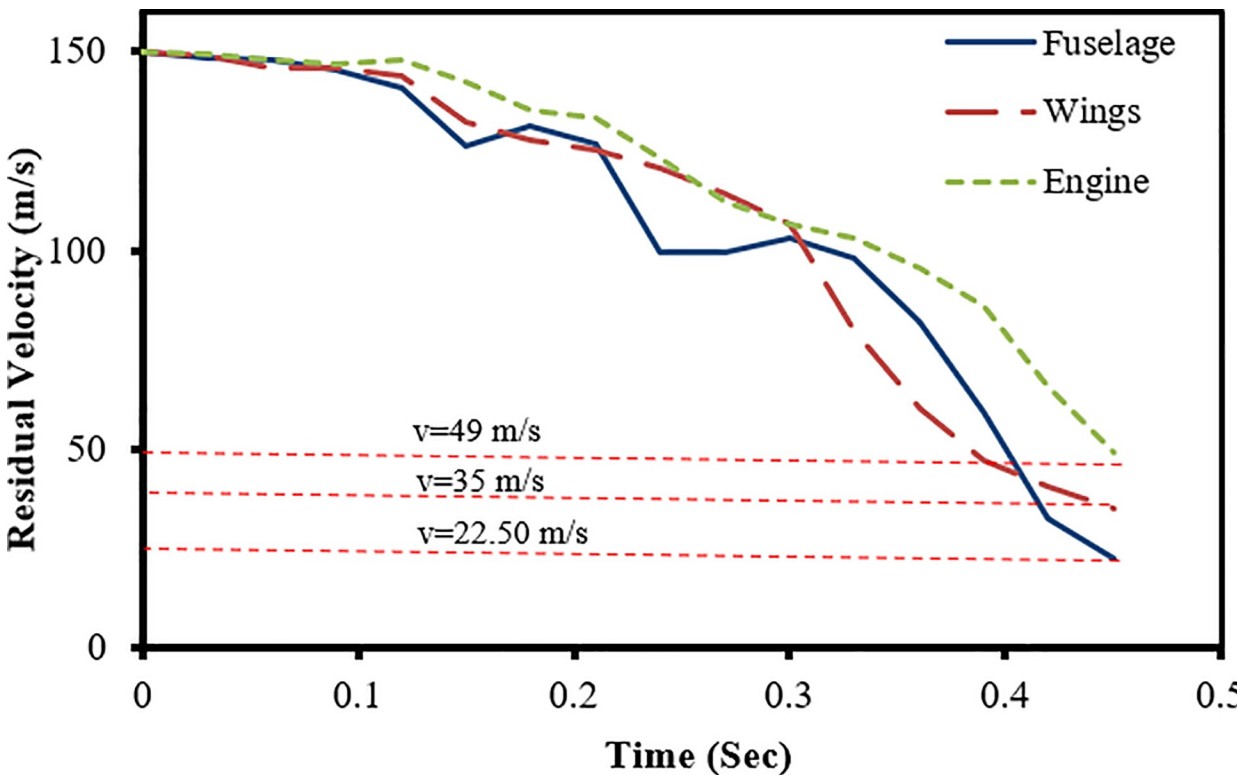

**Fig 17. Residual velocity of uncrushed aircraft parts against 150 m/s impact speed of aircraft.**

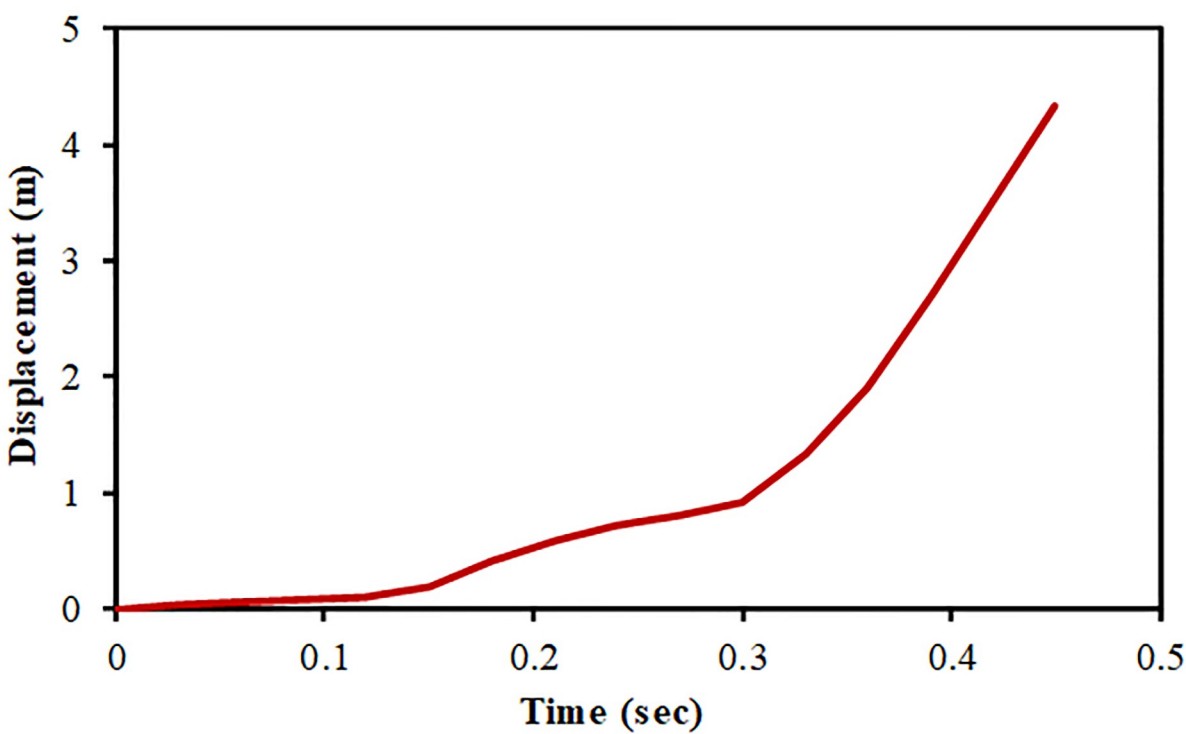

**Fig 18. Displacement of node 49625 at impact area against 150 m/s speed.**

Koshika [28] and impact tests simulation results reported by the author in a previous work [16].

### Force time-history analysis method

The impact analysis is also performed by force time history analysis method with force history developed by Riera method. This is accomplished by the application of the force time histories corresponding to 100 m/s and 150 m/s impact velocities at impact area shown in Fig 7. It is worth mentioning that results regarding residual velocities, kinetic energies, and fracture process of aircraft and target structures cannot be obtained in force time history method due to its limitations. Further, it can only be applied normal at the target as mentioned in assumptions of Riera method. On the other hand, the said responses can be obtained in missile target interaction method and aircraft can be oriented at an arbitrary/desired angle. In this analysis, the impact area is assumed based on impact of Boeing 767 on outer containment building which is sensitive to the response of the target structure [5]. Accordingly, this area is determined carefully by selecting elements in loading areas so that accurate representation of Riera force history is achieved as illustrated in Fig 7.

For full-scale impact analysis of aircraft force time history corresponding to 100 m/s, contours of maximum displacement and displacement time history at node 133 are shown in Fig 19. This node is located at center of the impact area on outer containment building and represents maximum displacement. It is observed that displacement of 0.3 m at 0.4 sec occurs which is less than the thickness of containment wall (0.9 m). From displacement contour and time history, it can be inferred that deformation of 0.3 m has occurred while no perforation or scabbing is observed. At impact time of 0.4 sec corresponding to maximum displacement, maximum principal stress and axial stresses in concrete and steel rebars are shown in Figs 20

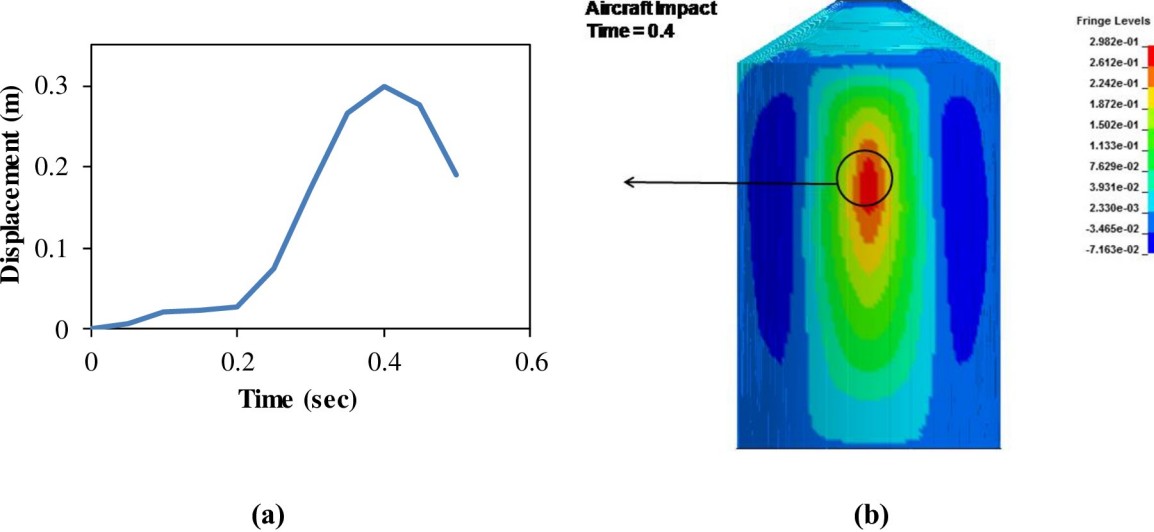

**Fig 19. Displacement time history and contours of max displacement for force time history corresponding to 100 m/s Riera curve.** (a) Maximum displacement time history. (b) Contours of max displacement.

and 21, respectively. These contours indicate that elements in red color reached the yield strength value defined for rebars, but no significant cracking and damage is observed in both concrete and steel rebars. From the results at impact velocity of 100 m/s, it can be found that

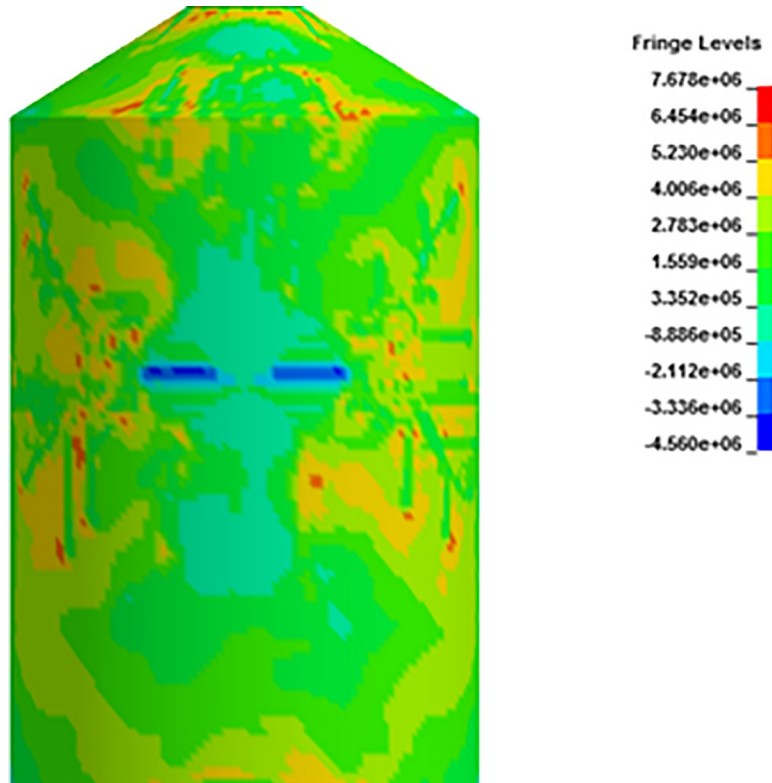

**Fig 20. Contour of maximum principal stress for force time history corresponding to 100 m/s impact speed curve.**

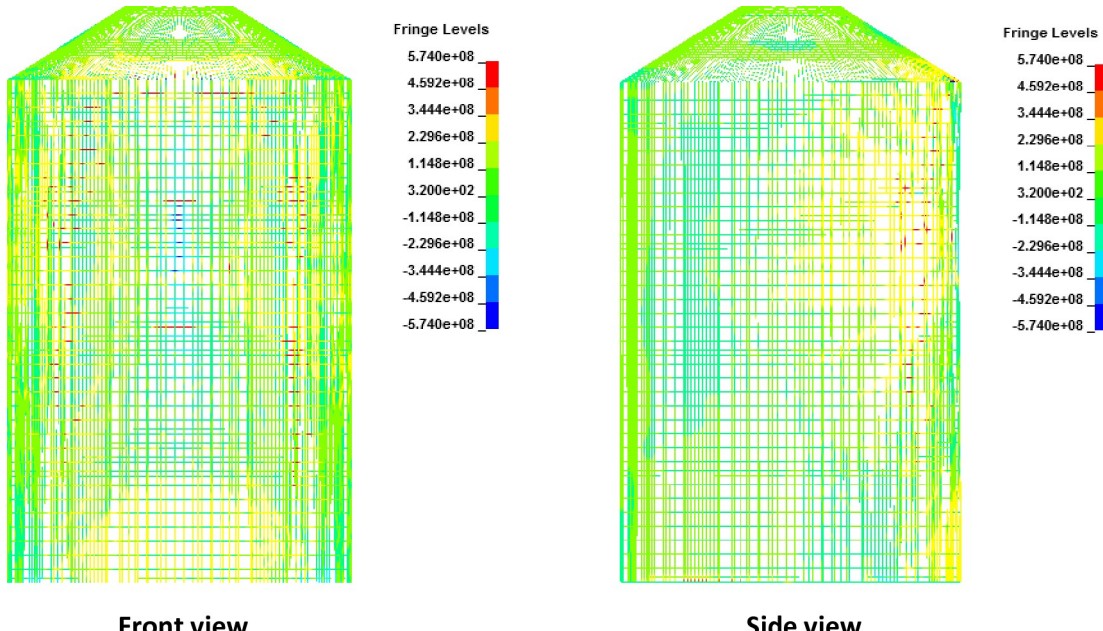

**Front view**                    **Side view**

**Fig 21.** Contours of axial stress for reinforcements against 100 m/s impact speed curve in front view (left) and side view (right).

no perforation and scabbing occurred in outer containment building and its overall integrity as a confinement barrier is ensured.

In analysis with force time history method, Riera curve corresponding to 150 m/s impact velocity is applied on loading area. Cracking is observed in outer containment building which finally leads to perforation as shown in Figs 22–24. Contours of maximum displacement and nodal displacement history in Fig 22 indicate that maximum displacement of 7.10 m has occurred which is quite high compared with maximum displacement of 4 m found in missile

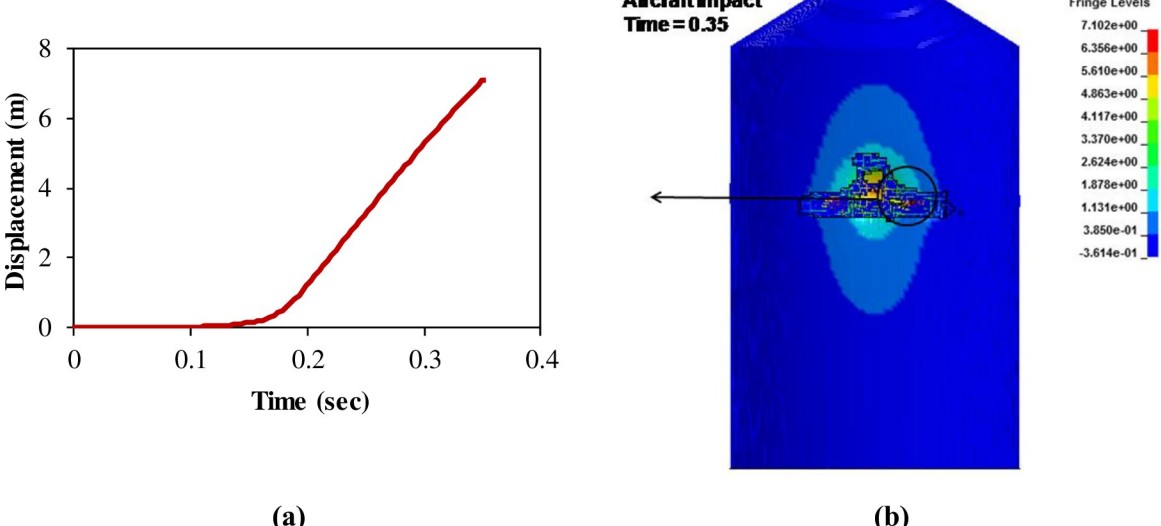

**(a)**                    **(b)**

**Fig 22. Displacement time history and contours of maximum displacement for force time history corresponding to 150 m/s Riera curve.** (a) Maximum displacement time history. (b) Contours of maximum displacement.

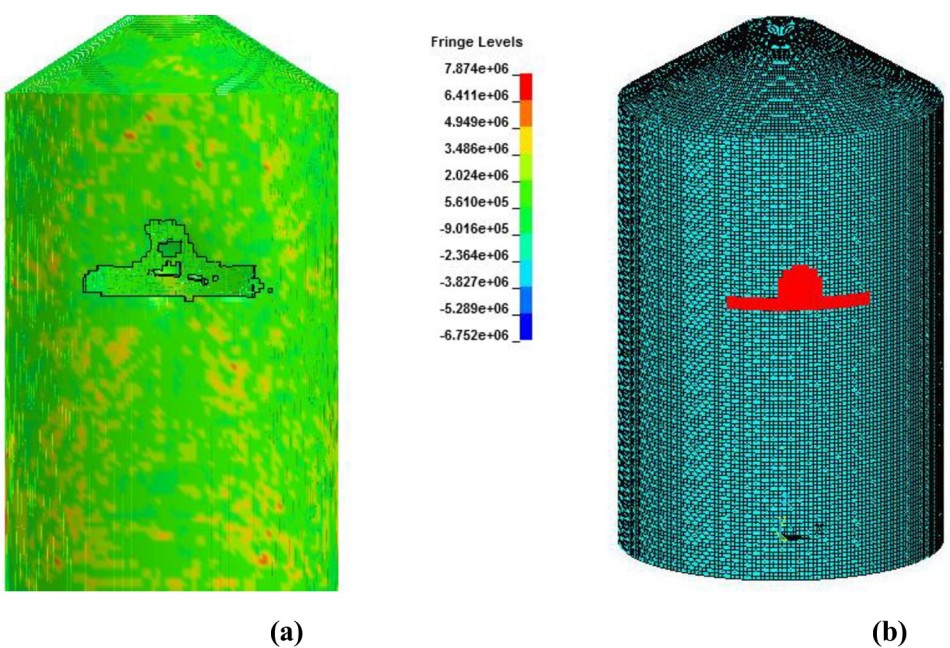

**Fig 23. Contours of max principal stress at 150 m/sec impact speed curve and comparison of damaged impact area vs. defined impact area.** (a) Damaged impact area. (b) Defined impact area.

target interaction method at 150 m/s. This may be due to conservative Riera curve and assumed loading area. A contour of maximum principal stress is given in Fig 23, illustrating the damage to concrete elements compared to defined elements at impact area for Riera force history. The damaged elements exhibit almost similar shape to that of defined area and verify

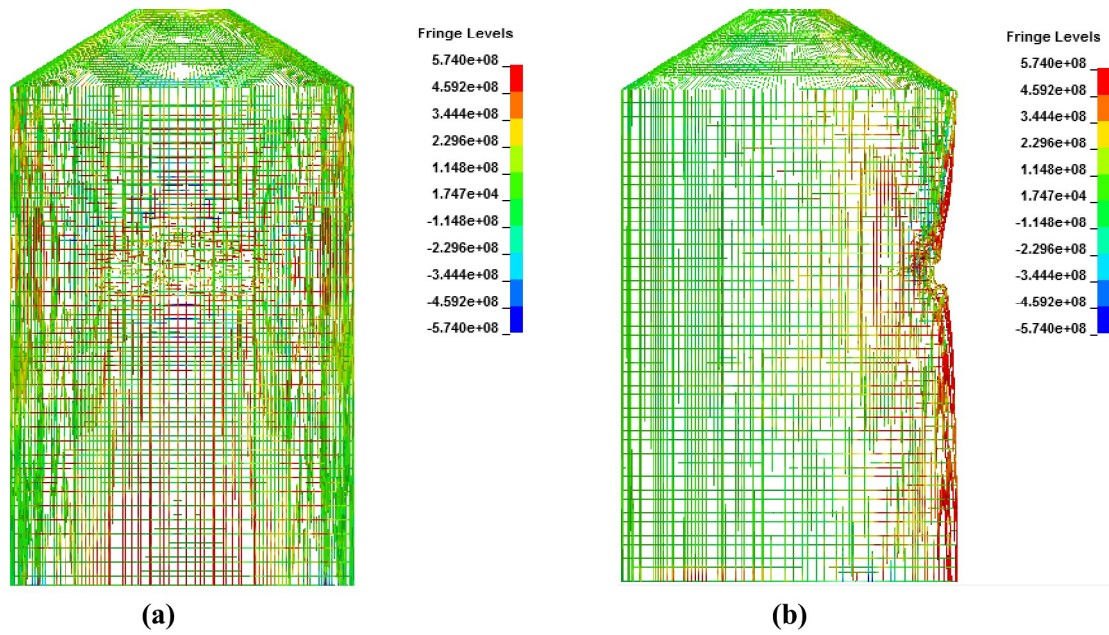

**Fig 24. Contours of axial stress for reinforcements at 150 m/s impact velocity in front view (left) and side view (right).** (a) Front view. (b) Side view.

the authenticity of force time history method. Contours of maximum axial stress in steel rebars are shown in Fig 24 which clearly indicate the breakage of steel rebars at impact area and higher axial stresses in surrounding reinforcements due to high velocity of aircraft and nonlinear nature of the impact.

## Comparison of results obtained from two methods

Force time history method is simplified one and, in general, overestimates the impact and impulse forces making this method conservative as reported in NEI 07–13 [4] and other published works [5, 6, 26]. Similarly, in current rigid wall impact analysis, the normal impact forces calculated by Riera method both for α = 1 and α = 0.9 are more than corresponding impact forces obtained in LS DYNA run as shown in Table 3. Accordingly, the results mentioned in above sections for force time history method are higher than missile target interaction method. The later method is more complex as it considers the detail modelling of aircraft & target structure, associated non linearities especially the interaction between aircraft & target as contact problem. Therefore, it gives more accurate and realistic results provided that sufficient aircraft data is available (especially exact mass and initial impact velocity). In below sections, comparison of some critical parameters obtained by both methods is presented.

### Comparison for 100 m/s impact velocity case

During the full-scale aircraft impact analysis for both methods, no local effects in terms of perforation and scabbing of concrete are observed. From Fig 25, it is evident that more elements are in tension in case of force time history method than that of missile target interaction method at impact areas. In Fig 26, comparison of contours of minimum principal stress (tensile stress) for concrete are presented. The maximum tensile stresses observed are 3.81 MPa and 2.15 MPa for force time history and missile target interaction methods, respectively. Both stresses are taken

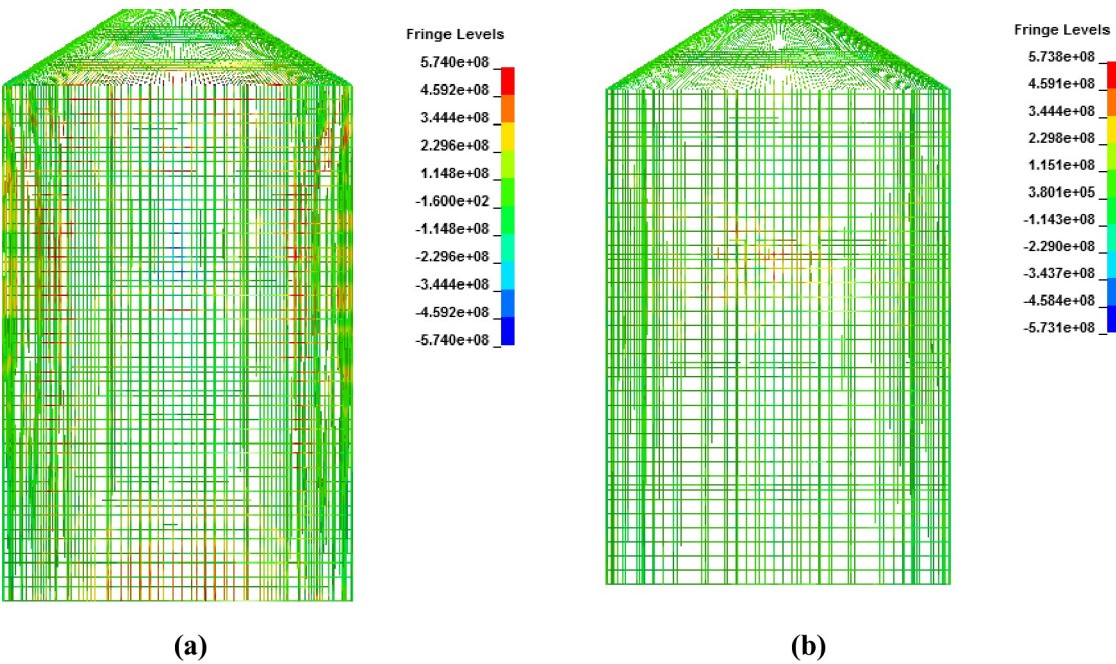

**(a)** **(b)**

**Fig 25. Comparison of axial stresses between two methods for vertical and hoop reinforcements at the end of impact.** (a) Force time history method. (b) Missile-target Interaction method.

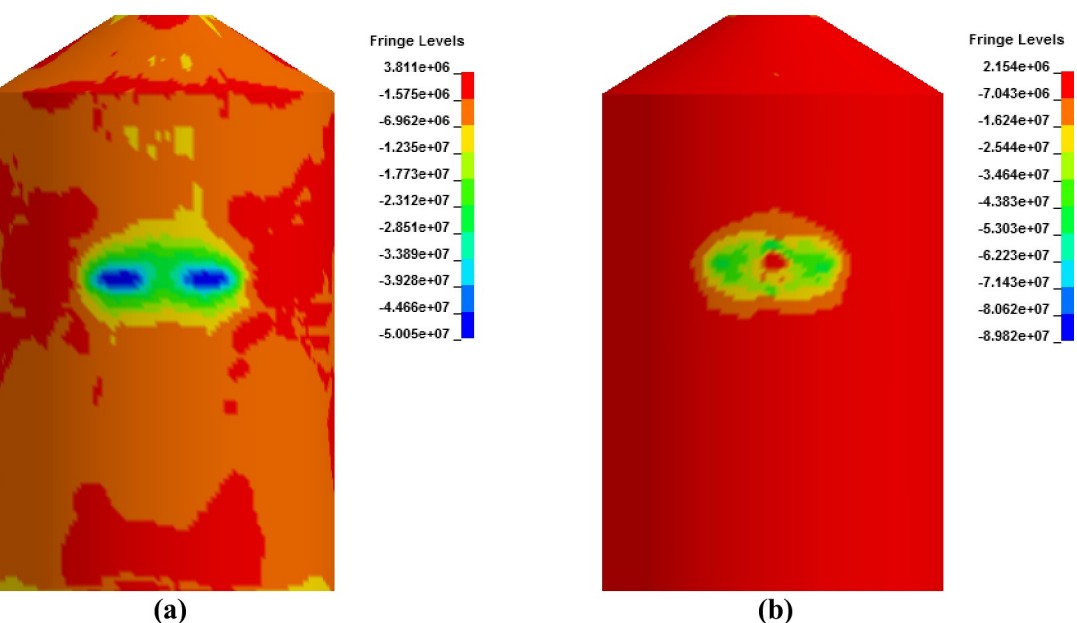

**Fig 26. Comparison of minimum principal stress (called tensile stresses for concrete) between two methods at 0.24 sec.** (a) Force time history method. (b) Missile-target interaction method.

corresponding to maximum impact forces mentioned in Fig 11(A). Although, tensile stress for concrete in force time method is more than missile target interaction method, yet these values are lower than allowable tensile stress value of 6.18 MPa defined in material model for concrete. This indicates that at 100 m/s impact velocity, no cracking is observed in concrete.

## Comparison for 150 m/s impact velocity case

In Fig 27, damaged areas of deleted elements reported in LS-DYNA post processor (LS-PrePost) are compared for both missile target interaction and force time history methods. The sizes of damaged elements are approximately 22.42 m × 10 m and 15.67 m × 10 m for force time history method and missile target interaction method, respectively. It is found that damage in case of missile target interaction method is caused by wings, fuel and engine of the aircraft. This is augmented by the observation that in simulation analysis with both steel plate reinforced concrete (SC) and reinforced concrete (RC) slab panels, damage (in terms of local effects) is caused after engine comes into contact with target [16]. While, in case of force time history method, Riera curve is applied based on assumed wings and fuselage areas. Therefore, in both methods, damaged areas represent different pattern which seems to be logical. In Fig 28, more elements are under axial stresses in case of force time history method than that of missile target interaction method. In failure criteria of concrete with *MAT_084, element erosion is activated by defining the maximum principal strain of concrete as 0.05. Accordingly, the maximum principal strain as result of full-scale impact analysis by both methods is illustrated in Fig 29. In analytical simulations presented, the elements eroded/spalled once they reached the predefined value of principal strains. More elements are deleted in the case of force time history method.

## Conclusions

Impact analysis of Boeing 767 aircraft is performed against the reinforced concrete outer containment building in ANSYS/LS-DYNA with two methods, namely, force time history and

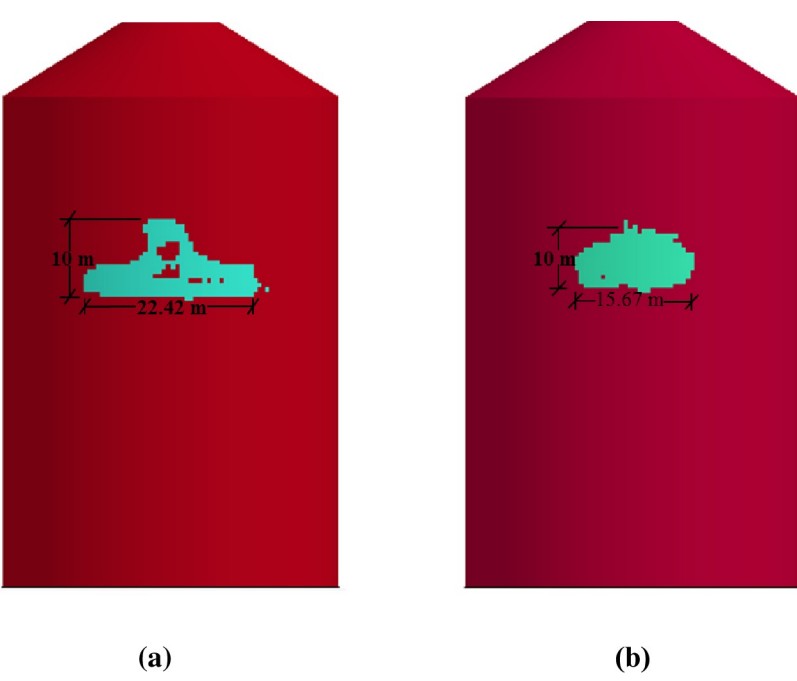

**(a)** **(b)**

**Fig 27. Comparison of damage to front side of outer containment building, at the end of impact.** (a) Force time history method. (b) Missile-target Interaction method.

missile target interaction methods. Defined FE model of aircraft is validated and full-scale aircraft impact analysis is performed to understand the behavior of both aircraft and target structure. The results in terms of fracture process, displacement time histories, maximum and minimum principal stresses, axial stresses of reinforcements and size of damaged area are

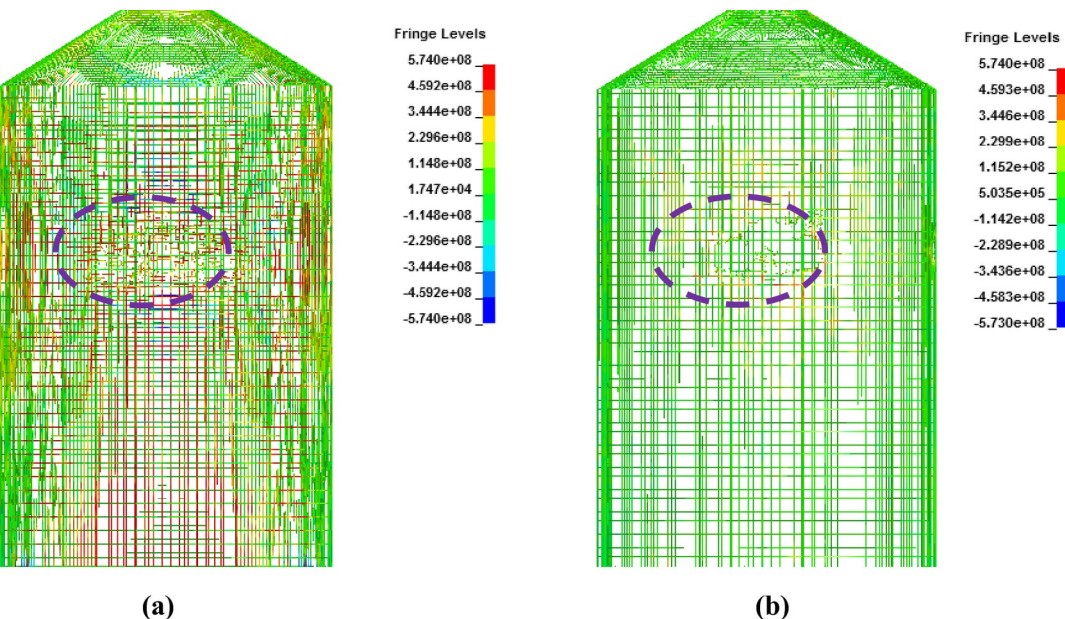

**(a)** **(b)**

**Fig 28. Comparison of axial stresses for reinforcements between two methods, at the end of impact.** (a) Force time history method. (b) Missile-target interaction method.

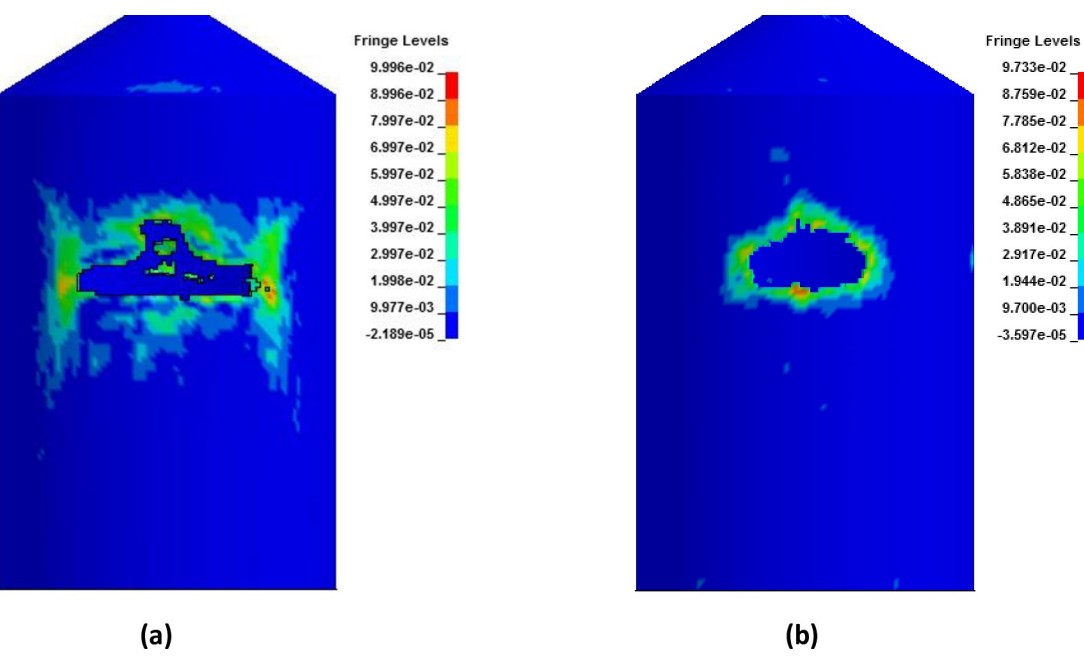

**(a)**                                                        **(b)**

**Fig 29. Comparison of maximum principal strain in concrete observed in two methods, at the end of impact.** (a) Force time history method (b) Missile-target interaction method.

presented and compared between two methods. The validation and comparison of results is useful to instill confidence in presented methodology and provide further guidelines for the applicability in specific applications.

The results reveal that impact loads (forces and impulses), displacements, stresses for concrete and steel reinforcement, and number of damaged elements are higher in case of force time history method than missile target interaction method, making the former one relatively conservative. Meanwhile, missile target interaction method is more realistic and in present study, addresses the limitations of force time history method in terms of fracture process, residual velocities and contact nonlinearity. During the full-scale impact analysis, no perforation or scabbing is observed in case of 100 m/s initial impact speed, thus preventing any potential leakage. With full mass of Boeing 767 and initial impact speed around 100 m/s, the overall integrity of outer containment building of NPPs, in present simulation, is assured and its function as a confinement barrier is guaranteed. At speeds higher than 120 m/s, local failure modes like scabbing, penetration and perforation are dominant and there are chances of radiation leakage from annular space. The results illustrate the increase in damage trends of outer containment with increasing impact speed. This concludes that in design and assessment of important structures like outer containment building against aircraft loadings, sufficient thickness of RC cylindrical wall & dome or consideration of steel plates are essential to account for local failure modes and overall structural integrity.

In rigid wall impact analysis, the impact forces from Riera method and finite element method are comparable which satisfy the recommendations set forth by NEI 07–13 [4] and further ensure the accuracy of results in full-scale impact analysis. By present simulation results for both methods, it is found that Winfrith concrete model (*MAT_084) can also simulate the nonlinear response of concrete in full-scale impact of aircraft on reinforced concrete containment and endorses the conclusions of previously published study on simulation of scale model tests [16]. Based on this study, it is recommended that internal structures of an

aircraft should also be simulated simultaneously to judge their behavior and get more realistic results. The whole methodology presented in this paper may be adopted for future studies related to full-scale aircraft impact on important facilities like nuclear power plants.

## Acknowledgments

The Authors would like to thank Dr. Pan Rong (NSC) for providing opportunity to perform initial work during joint research project.

## Author Contributions

**Conceptualization:** Rao Arsalan Khushnood, Wasim Khaliq.

**Data curation:** Muhammad Sadiq.

**Formal analysis:** Muhammad Sadiq.

**Investigation:** Muhammad Sadiq.

**Methodology:** Muhammad Sadiq, Rao Arsalan Khushnood, Wasim Khaliq.

**Project administration:** Rao Arsalan Khushnood, Wasim Khaliq.

**Resources:** Rao Arsalan Khushnood.

**Software:** Pan Rong.

**Supervision:** Rao Arsalan Khushnood, Wasim Khaliq.

**Validation:** Rao Arsalan Khushnood, Muhammad Ilyas, Wasim Khaliq, Shaukat Ali Khan.

**Visualization:** Muhammad Ilyas.

**Writing – original draft:** Muhammad Sadiq.

**Writing – review & editing:** Rao Arsalan Khushnood, Muhammad Ilyas, Wasim Khaliq, Shaukat Ali Khan, Pan Rong.

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
