## [Decision Letter · Decision Letter 0]

27 Apr 2020

PONE-D-20-08713

Comparative assessment of impact analysis methods applied to large commercial aircraft crash on reinforced concrete containment

PLOS ONE

Dear Dr. Khushnood,

Thank you for submitting your manuscript to PLOS ONE. After careful consideration, we feel that it has merit but does not fully meet PLOS ONE’s publication criteria as it currently stands. Therefore, we invite you to submit a revised version of the manuscript that addresses the points raised during the review process.

Please address the comments from reviewers.

We would appreciate receiving your revised manuscript by Jun 11 2020 11:59PM. To enhance the reproducibility of your results, we recommend that if applicable you deposit your laboratory protocols in protocols.io, where a protocol can be assigned its own identifier (DOI) such that it can be cited independently in the future. For instructions see: http://journals.plos.org/plosone/s/submission-guidelines#loc-laboratory-protocols

We look forward to receiving your revised manuscript.

Kind regards,

Jianguo Wang, PhD

Academic Editor

PLOS ONE

Journal Requirements:

'The research presented in this paper was supported by National University of Sciences and Technology (NUST), Islamabad, Pakistan...'

'The author(s) received no specific funding for this work.'

Reviewers' comments:

Reviewer's Responses to Questions

**Comments to the Author**

1. Is the manuscript technically sound, and do the data support the conclusions?

Reviewer #1: Partly

Reviewer #2: Yes

2. Has the statistical analysis been performed appropriately and rigorously? 

Reviewer #1: Yes

Reviewer #2: Yes

3. Have the authors made all data underlying the findings in their manuscript fully available?

Reviewer #1: Yes

Reviewer #2: Yes

4. Is the manuscript presented in an intelligible fashion and written in standard English?

Reviewer #1: Yes

Reviewer #2: Yes

5. Review Comments to the Author

Reviewer #1: In this paper, a comparative analysis is made between force time history method and missile target interaction method for damage assessment of a large aircraft impacting reinforced concrete containment. An aircraft of Boeing 767 is greatly simplified to develop finite element model, and comparisons of impact force obtained by impact of the finite element model on the rigid wall and the Riera method are performed to verify rationality of the aircraft model. Finally, the damage of reinforced concrete containment is analyzed by two methods, including displacement，stress for concrete and steel reinforcement，and number of damaged elements. The paper is worth publishing. However, the following concerns need to be addressed in revising the manuscript.

Reviewer #2: The precise evaluation of the potential damage caused by large commercial aircraft crash into civil structures has been the focus in the field of explosion and shock. The results in terms of fracture process, displacement time histories, maximum and minimum principal stresses, axial stresses of reinforcements and size of damaged area are presented and compared between two methods in this paper, which has important reference value. Besides, the idea of the article is clear and the language is concise.

6. PLOS authors have the option to publish the peer review history of their article (what does this mean?). If published, this will include your full peer review and any attached files.

Reviewer #1: No

Reviewer #2: No

---

## [Author Response · Author response to Decision Letter 0]

30 Jun 2020

The authors appreciate the constructive suggestions by the worthy reviewers. We do agree with most of the reviewer’s suggestions and technical comments. The manuscript has been revised to incorporate all the suggestions of respected reviewers. The detailed response to each concern is included in the separate file attached with revised submission, please

---

## [Decision Letter · Decision Letter 1]

23 Jul 2020

Comparative assessment of impact analysis methods applied to large commercial aircraft crash on reinforced concrete containment

PONE-D-20-08713R1

Dear Dr. Khushnood,

We’re pleased to inform you that your manuscript has been judged scientifically suitable for publication and will be formally accepted for publication once it meets all outstanding technical requirements.

Kind regards,

Jianguo Wang, PhD

Academic Editor

PLOS ONE

Additional Editor Comments (optional):

Reviewers' comments:

Reviewer's Responses to Questions

**Comments to the Author**

1. If the authors have adequately addressed your comments raised in a previous round of review and you feel that this manuscript is now acceptable for publication, you may indicate that here to bypass the “Comments to the Author” section, enter your conflict of interest statement in the “Confidential to Editor” section, and submit your "Accept" recommendation.

Reviewer #1: All comments have been addressed

2. Is the manuscript technically sound, and do the data support the conclusions?

Reviewer #1: Yes

3. Has the statistical analysis been performed appropriately and rigorously? 

Reviewer #1: Yes

4. Have the authors made all data underlying the findings in their manuscript fully available?

Reviewer #1: Yes

5. Is the manuscript presented in an intelligible fashion and written in standard English?

Reviewer #1: Yes

6. Review Comments to the Author

Reviewer #1: (No Response)

7. PLOS authors have the option to publish the peer review history of their article (what does this mean?). If published, this will include your full peer review and any attached files.

Reviewer #1: No

---

## [Editor Report · Acceptance letter]

27 Jul 2020

PONE-D-20-08713R1 

Comparative assessment of impact analysis methods applied to large commercial aircraft crash on reinforced concrete containment 

Dear Dr. Khushnood:

I'm pleased to inform you that your manuscript has been deemed suitable for publication in PLOS ONE. Congratulations! Your manuscript is now with our production department. 

Kind regards, 

on behalf of

Dr. Jianguo Wang 

Academic Editor

PLOS ONE